# Better with Less: A Data-Active Perspective on Pre-Training Graph Neural Networks

**Jiarong Xu**[1*]**, Renhong Huang**[2]**, Xin Jiang**[3]**, Yuxuan Cao**[2]
**Carl Yang**[4]**, Chunping Wang**[5]**, Yang Yang**[2]
[1]Fudan University, [2]Zhejiang University, [3]Lehigh University
[4]Emory University, [5]Finvolution Group
jiarongxu@fudan.edu.cn, {renh2, caoyx, yangya}@zju.edu.cn
xjiang@lehigh.edu, j.carlyang@emory.edu, wangchunping02@xinye.com

## Abstract

Pre-training on graph neural networks (GNNs) aims to learn transferable knowledge for downstream tasks with unlabeled data, and it has recently become an active research area. The success of graph pre-training models is often attributed to the massive amount of input data. In this paper, however, we identify the *curse of big data* phenomenon in graph pre-training: more training data do not necessarily lead to better downstream performance. Motivated by this observation, we propose a *better-with-less* framework for graph pre-training: fewer, but carefully chosen data are fed into a GNN model to enhance pre-training. The proposed pre-training pipeline is called the data-active graph pre-training (APT) framework, and is composed of a graph selector and a pre-training model. The graph selector chooses the most representative and instructive data points based on the inherent properties of graphs as well as *predictive uncertainty*. The proposed predictive uncertainty, as feedback from the pre-training model, measures the confidence level of the model in the data. When fed with the chosen data, on the other hand, the pre-training model grasps an initial understanding of the new, unseen data, and at the same time attempts to remember the knowledge learned from previous data. Therefore, the integration and interaction between these two components form a unified framework (APT), in which graph pre-training is performed in a progressive and iterative way. Experiment results show that the proposed APT is able to obtain an efficient pre-training model with fewer training data and better downstream performance.

## 1 Introduction

Pre-training Graph neural networks (GNNs) shows great potential to be an attractive and competitive strategy for learning from graph data without costly labels [29, 50]. Recent advancements have been made in developing various graph pre-training strategies, which aim to capture transferable patterns from a diverse set of unlabeled graph data [22, 30, 31, 40, 44, 60, 76, 77]. Very often, the success of a graph pre-training model is attributed to the massive amount of unlabeled training data, a well-established consensus for pre-training in computer vision [12, 19, 26] and natural language processing [11, 45].

In view of this, contemporary research almost has no controversy on the following issue: *Is a massive amount of input data really necessary, or even beneficial, for pre-training GNNs?* Yet, two simple experiments regarding the number of training samples (and graph datasets) seem to doubt the positive

---

*Corresponding author.

37th Conference on Neural Information Processing Systems (NeurIPS 2023).

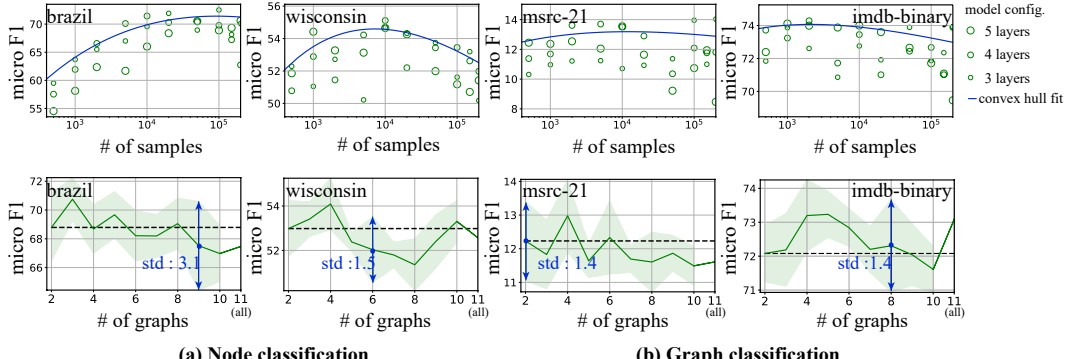

**(a) Node classification**    **(b) Graph classification**

Figure 1: *Top row*: The effect of scaling up sample size (log scale) on the downstream performance based on a group of GCCs [50] under different configurations (the graphs used for pre-training are kept as all eleven pre-training data in Table 3, and the samples are taken from the backbone pre-training model according to its sampling strategy). The results for different downstream graphs (and tasks) are presented in separate figures. To better show the changing trend, we fit a curve to the best performing models (*i.e.*, the convex hull fit as [1] does). *Bottom row*: The effect of scaling up the number of graph datasets on the downstream performance based on GCC. For a fixed horizontal coordinate, we run 5 trials. For each trial, we randomly choose a combination of input graphs. The shaded area indicates the standard deviation over the 5 trials. See Appendix E for more observations on other graph pre-training models and detailed settings.

answer to this question. We observe that scaling pre-training samples does not result in a one-model-fits-all increase in downstream performance (top row in Figure 1), and that adding input graphs (while fixing sample size) does not improve and sometimes even deteriorates the generalization of the pre-trained model (bottom row in Figure 1). Moreover, even if the number of input graphs (the horizontal coordinate) is fixed, the performance of the model pre-trained on different combinations of inputs varies dramatically; see the standard deviation in blue in Figure 1. As the first contribution of this work, we identify the *curse of big data* phenomenon in graph pre-training: more training samples (and graph datasets) do not necessarily lead to better downstream performance.

Therefore, instead of training on a massive amount of data, it is more appealing to choose a few suitable samples and graphs for pre-training. However, without the knowledge of downstream tasks, it is difficult to design new data selection criteria for the pre-training model. To this end, we propose the *graph selector* which is able to provide the most instructive data for the model by incorporating two criteria: *predictive uncertainty* and *graph properties*. On one hand, predictive uncertainty is introduced to measure the level of confidence (or certainty) in the data. On the other hand, some graphs are inherintly more informative and representative than others, and thus the fundamental properties should help in the selection process.

Apart from the graph selector, the pre-training model is also designed to co-evolve with the data. Instead of swallowing data as a whole, the pre-training model is encouraged to learn from the data in a progressive way. After learning a certain amount of training data, the model receives feedback (from predictive uncertainty) on what kind of data the model has least knowledge of. Then the pre-training model is able to reinforce itself on highly uncertain data in next training iterations.

Putting together, we integrate the graph selector and the pre-training model into a unified paradigm and propose a data-active graph pre-training (APT) framework. The term "data-active " is used to emphasize the co-evolution of data and model, rather than mere data selection before model training. The two components in the framework actively cooperate with each other. The graph selector recognizes the most instructive data for the model; equipped with this intelligent selector, the pre-training model is well-trained and in turn provides better guidance for the graph selector.

The rest of the paper is organized as follows. In §2 we present the basic graph pre-training framework and review existing work on this topic. Then in §3 we describe in detail the proposed data-active graph pre-training (APT) paradigm. §4 contains numerical experiments, demonstrating the superiority of APT in different downstream tasks, and also includes the applicable scope of our pre-trained model.

## 2 Basic Graph Pre-training Framework

This section reviews the basic framework of graph pre-training commonly used in the literature. The backbone of our graph pre-training model also follows this framework.

We start with a natural question: What does graph pre-training actually learn? On one hand, graph pre-training tries to learn transferable semantic meaning associated with structural patterns. For example, both in citation networks and social networks, the closed triangle structure (▨) is interpreted as a stable relationship, while the open triangle (▧) indicates an unstable relationship. In comparison, these semantic meanings can be quite different in other networks, *e.g.*, molecular networks. On the other hand, however, the distinction (or relationship) between different structural patterns is still transferable. Taking the same example, the closed and open triangles might yield different interpretations in molecular networks (stability of certain chemical property) from those in social networks (stability in social intimacy), but the distinction between these two structures remains the same: they indicate opposite (or contrastive) semantic meanings [32, 41, 57]. Therefore, the graph pre-training either learns representative structural patterns (when semantic meanings are present), or more importantly, obtains the capability of distinguishing these patterns. This observation in graph pre-training is not only different from that in other areas (*e.g.*, computer vision and natural language processing), but may also explain why graph pre-training is effective, especially when some downstream information is absent.

With the hope to learn the transferable structural patterns or the ability to distinguish them, the graph pre-training model is fed with a diverse collection of input graphs, and the learned model, denoted by $f_\theta$ (or simply $f$ if the parameter $\theta$ is clear from context), maps a node to a low-dimensional representation. Unaware of the specific downstream task as well as task-specific labels, one typically designs a self-supervised task for the pre-training model. Such self-supervised information for a node is often hidden in its neighborhood pattern, and thus the structure of its ego network can be used as the transferable pattern. Naturally, subgraph instances sampled from the same ego network $\Gamma_i$ are considered *similar* while those sampled from different ego networks are rendered *dissimilar*. Therefore, the pre-training model attempts to capture the similarities (and dissimilarities) between subgraph instances, and such a self-supervised task is called the *subgraph instance discrimination task*. More specifically, given a subgraph instance $\zeta_i$ from an ego network $\Gamma_i$ centered at node $v_i$ as well as its representation $\boldsymbol{x}_i = f(\zeta_i)$, the model $f$ aims to encourage higher similarity between $\boldsymbol{x}_i$ and the representation of another subgraph instance $\zeta_i^+$ sampled from the same ego network. This can be done by minimizing, *e.g.*, the InfoNCE loss [48],

$$\mathcal{L}_i = -\log \frac{\exp\left(\boldsymbol{x}_i^\top f(\zeta_i^+)/\tau\right)}{\exp\left(\boldsymbol{x}_i^\top f(\zeta_i^+)/\tau\right) + \sum\limits_{\zeta_i' \in \Omega_i^-} \exp\left(\boldsymbol{x}_i^\top f(\zeta_i')/\tau\right)}, \tag{1}$$

where $\Omega_i^-$ is a collection of subgraph instances sampled from different ego networks $\Gamma_j$ ($j \neq i$), and $\tau > 0$ is a temperature hyper-parameter. Here the inner product is used as a similarity measure between two instances. One common strategy to sample these subgraph instances is via random walks on graphs, as used in GCC [50], and other sampling methods as well as loss functions are also valid.

## 3 Data-Active Graph Pre-training

In this section, we present the proposed APT framework for graph pre-training, and the overall pipeline is illustrated in Figure 2. The APT framework consists of two major components: a graph selector and a graph pre-training model. The technical core is the interaction between these two components: The graph selector feeds *suitable* data for pre-training, and the graph pre-training model learns from the carefully chosen data. The feedback of the pre-training model in turn helps select the needed data tailored to the model.

The rest of this section is organized as follows. We describe the graph selector in §3.1 and the graph pre-training model in §3.2. The overall pre-training and fine-tuning strategy is presented in §3.3.

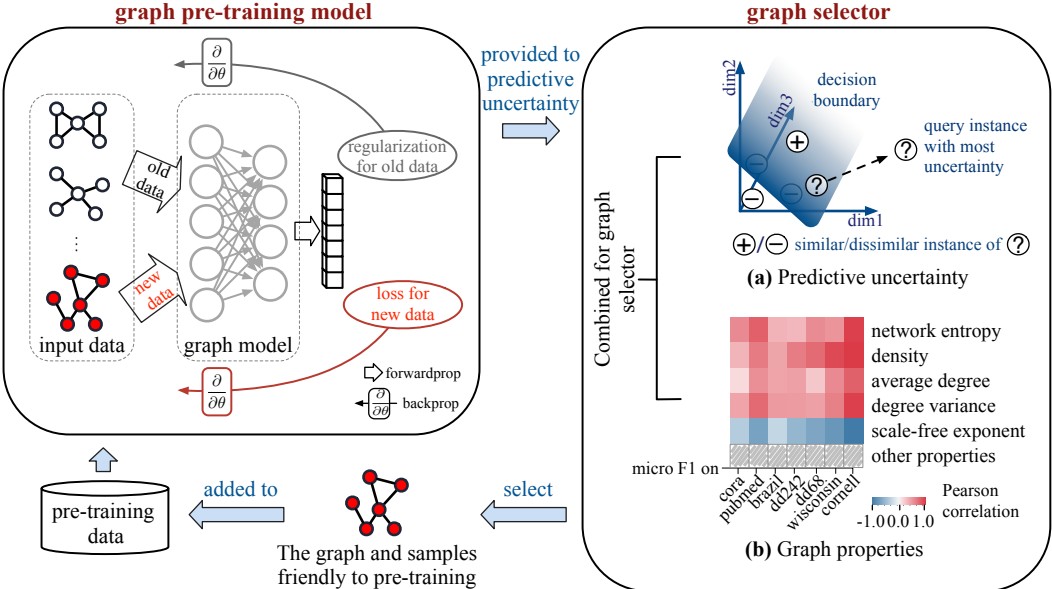

Figure 2: Overview of the proposed data-active graph pre-training paradigm. The graph selector provides the graph and samples suitable for pre-training, while the graph pre-training model learns from the incoming data in a progressive way and in turn better guides the selection process. In the graph selector component, *Part (a)* provides an illustrating example on the predictive uncertainty, and *Part (b)* plots the Pearson correlation between the properties of the input graph and the performance of the pre-trained model on the training set (using this graph) when applied to different unseen test datasets (see Appendix F for other properties that exhibit little/correlation with performance).

## 3.1 Graph selector

In view of the curse of big data phenomenon, it is more appealing to carefully choose data *well suited* for graph pre-training rather than training on a massive amount of data. Conventionally, the criterion of suitable data, or the contribution of a data point to the model, is defined based on the output predictions on downstream tasks [20]. In graph pre-training where downstream information is absent, new selection criteria or guidelines are needed to provide effective instructions for the model. Here we introduce two kinds of selection criteria, originating from different points of view, to help select suitable data for pre-training. The *predictive uncertainty* measures the model's understanding of certain data, and thus helps select the least certain data points for the current model. In addition to the measure of model's ability, some *inherent properties* of graphs can also be used to assess the level of representativeness or informativeness of a given graph.

**Predictive uncertainty.** The notion of predictive uncertainty can be explained via an illustrative example, as shown in part (a) of the graph selector component in Figure 2. Consider a query subgraph instance $\zeta_i$ (denote by ⑦ in Figure 2) from the ego network $\Gamma_i$ in a graph $G$. If the pre-training model cannot tell its similar instance $\zeta_i^+$ (denoted by ⊕) from its dissimilar instance $\zeta_i^- \in \Omega_i^-$ (denoted by ⊖), we say that the current model is uncertain about the query instance $\zeta_i$. Therefore, the contrastive loss function in Eq. (1) comes in handy as a natural measure for the predictive uncertainty of the instance $\zeta_i$: $\phi_{\text{uncertain}}(\zeta_i) = \mathcal{L}_i$. Accordingly, the predictive uncertainty of a graph $G$ (*i.e.*, the graph-level predictive uncertainty) is defined as $\phi_{\text{uncertain}}(G) = (1/M) \sum_{i=1}^{M} \mathcal{L}_i$, where $M$ is the number of subgraph instances queried in this graph.

We further establish a provable connection between the proposed predictive uncertainty and the conventional definition of uncertainty. In most existing work, model uncertainty is often defined in the label space, *e.g.*, taking as the uncertainty the cross entropy loss $\mathcal{L}_{\text{CE}}(\zeta)$ of an instance $\zeta$ on the downstream classifier [43, 54, 56, 58]. Comparatively, our definition of uncertainty, $\phi_{\text{uncertain}}(\zeta)$, is in the representation space. The theoretical connection between these two losses is given in the following theorem.

**Theorem 1** (Connection between uncertainties.)**.** *Let $\mathcal{X}$, $\mathcal{Z}$ and $\mathcal{Y}$ be the input space, representation space and label set of downstream classifier. Denote a downstream classifier by $h: \mathcal{Z} \to \mathcal{Y}$ and the*

*set of downstream classifiers by $\mathcal{H}$. Assume that the distribution of labels is a uniform distribution over $\mathcal{Y}$. For any graph encoder $f : \mathcal{X} \to \mathcal{Z}$, one has*

$$\mathcal{L}_{\text{CE}}(\mathcal{X}) \geq \log\left(\frac{\log|\mathcal{Y}|}{\log 2 - \phi_{\text{uncertain}}(\mathcal{X})}\right),$$

*where $\mathcal{L}_{\text{CE}}$ denotes the conventional uncertainty, defined as cross entropy loss and estimated from the composition of graph encoder and downstream classifier $h \circ f$, and $\phi_{\text{uncertain}}$ is the proposed uncertainty estimated from graph encoder $f$ (independent of the downstream classifier).*

While the proof and additional discussion on the advantage of $\phi_{\text{uncertain}}$ are postponed to Appendix B, we emphasize here that, by Theorem 1, a smaller $\mathcal{L}_{\text{CE}}$ over all downstream classifiers cannot be achieved without a smaller $\phi_{\text{uncertain}}$.

Although GCC is used as the backbone model in the presented framework, our data selection strategy can be easily adapted to other non-contrastive learning tasks. In that case, the InfoNCE loss used in $\phi_{\text{uncertain}}$ should be replaced with another pre-training loss associated with the specific learning task. More details with the example of graph reconstruction is included in Appendix H.

**Graph properties.** As we see above, the predictive uncertainty measures the model's ability to distinguish (or identify) a given graph (or subgraph instance). However, predictive uncertainty is sometimes misleading, especially when the chosen graph (or subgraph) happens to be an outlier of the entire data collection. Hence, learning solely from the most uncertain data is not enough to boost the overall performance, or worse still, might lead to overfitting. The inherent properties of the graph turn out to be equivalently important as a selection criterion for graph pre-training. Intuitively, it is preferable to choose those graphs that are *good by themselves*, *e.g.*, those with better structure, or those containing more information. To this end, we introduce five inherent properties of graphs, *i.e.*, network entropy, density, average degree, degree variance and scale-free exponent, to help select *better* data points for pre-training. All these properties exhibit a strong correlation with downstream performance, which is empirically verified and presented in part (b) of the graph selector component in Figure 2. The choice of these properties also has an intuitive explanation, and here we discuss the intuition behind the network entropy as an example.

The use of *network entropy* is inspired from the sampling methods used in many graph pre-training models (see, *e.g.*, [22, 50]).In those works, random walks are used to construct subgraph instances (used as model input). Random walks can can also serve as a means to quantify the amount of information contained in a graph. In particular, the amount of information contained in a random walk from node $v_i$ to node $v_j$ is defined as $-\log P_{ij}$ [6], where $P$ is the transition matrix. Thus, the network entropy of a connected graph can be defined as the expected information of individual transitions over the random walk process [3]:

$$\phi_{\text{entropy}} = \langle -\log P\rangle_P = -\sum_{i,j} \pi_i P_{ij} \log P_{ij}, \quad (2)$$

where $\boldsymbol{\pi}$ is the stationary distribution of the random

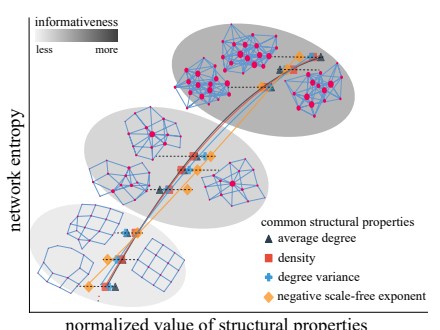

Figure 3: Illustrative graphs with increasing network entropy (bottom left to top right), and the other four graph properties.

walk and $\langle \cdot \rangle_P$ denotes the expectation of a random variable according to $P$. Network entropy (2) is in general difficult to calculate, but is still tractable for special choices of the transition matrix $P$. For example, for a connected unweighted graph $G = (V, E)$ with node degree vector $\boldsymbol{d} \in \mathbb{R}^{|V|}$, if the transition matrix is defined by $P_{ij} = 1/d_i$, then the stationary distribution is $\boldsymbol{\pi} = (1/2|E|)\boldsymbol{d}$ and the network entropy (2) reduces to

$$\phi_{\text{entropy}} = \frac{1}{2|E|} \sum_{i=1}^{|V|} d_i \log d_i. \quad (3)$$

In this case, the network entropy of a graph depends solely on its degree distribution, which is straightforward and inexpensive to compute.

Although the definition of network entropy originates from random walks on graphs, it is still useful in graph pre-training even when the sampling of subgraph instances does not depend on random

walks. Its usefulness can also be explained via the coding theory. Network entropy can be viewed as the entropy rate of a random walk, and it is known that the entropy rate is the expected number of bits per symbol required to describe a stochastic process [6]. Similarly, the network entropy can be interpreted as the expected number of "words" needed to describe the graph. Thus, intuitively, the larger the network entropy is, the more information the graph contains.

We also note that the connectivity assumption does not limit the usefulness of Eq. (3) in our case. For disconnected input graphs, we can simply compute the network entropy of the largest connected component, since for most real-world networks, the largest connected component contains most of the information [14]. Alternatively, we can also take some of the largest connected components from the graph and treat them separately as several connected graphs.

Furthermore, the other four graph properties, *i.e.*, density, average degree, degree variance and scale-free exponent, are closely related to the network entropy. Figure 3 presents a clear correlation between the network entropy and the other four graph properties, as well as provides some illustrative graphs. (These example graphs are generated by the configuration model proposed in [47], and Appendix F contains more results on real-world networks.) Intuitively, graphs with higher network entropy contain a larger amount of information, and so are graphs with larger density, higher average degree, higher degree variance, or a smaller scale-free exponent. The connections between all five graph properties can also be theoretically justified and the motivations of choosing these properties can be found in Appendix A. The detailed empirical justification of these properties and the pre-training performance in included in Appendix F.

**Time-adaptive selection strategy.** The proposed predictive uncertainty and the five graph properties together act as a powerful indicator of a graph's suitability for a pre-training model. Then, we aim to select the graph with the highest score, where the score is defined as

$$\mathcal{J}(G) = (1 - \gamma_t)\hat{\phi}_{\text{uncertain}} + \gamma_t\text{MEAN}(\hat{\phi}_{\text{entropy}}, \hat{\phi}_{\text{density}}, \hat{\phi}_{\text{avg\_deg}}, \hat{\phi}_{\text{deg\_var}}, \text{-}\hat{\phi}_\alpha), \quad (4)$$

where the optimization variable is the graph $G$ to be selected, $\gamma_t \in [0, 1]$ is a trade-off parameter to balance the weight between predictive uncertainty and graph properties, and $t$ is the iteration counter. The small hat on the terms $\hat{\phi}_{\text{uncertain}}$, $\hat{\phi}_{\text{entropy}}$, $\hat{\phi}_{\text{density}}$, $\hat{\phi}_{\text{avg\_deg}}$, $\hat{\phi}_{\text{deg\_var}}$ and $\hat{\phi}_\alpha$ indicates that all the values are already $z$-normalized, so the objective (especially the MEAN operator) is independent of their original scales.

Note that the pre-training model learns nothing at the beginning, so we initialize $\gamma_0 = 1$, and in later iterations, the balance between the predictive uncertainty and the inherent graph properties ensures that the selected graph is a good supplement to the current pre-training model as well as an effective representative for the entire data distribution. In particular, at the beginning of the pre-training, the outputs of the model are not accurate enough to guide data selection, so the parameter $\gamma_t$ should be set larger so that the graph properties play a leading role. As the training phase proceeds, the graph selector gradually pays more attention to the feedback $\phi_{\text{uncertain}}$ via a smaller value of $\gamma_t$. Therefore, in our framework, the parameter $\gamma_t$ is called the *time-adaptive parameter*, and is set to be a random variable depending on time $t$. In this work, we take $\gamma_t$ from a Beta distribution $\gamma_t \sim \text{Beta}(1, \beta_t)$, where $\beta_t$ decreases over time (training iterations).

## 3.2 Graph pre-training model

Instead of swallowing all the pre-training graphs as a whole, our graph pre-training model takes the input graphs and samples one by one in a sequential order and enhances itself in a progressive manner. However, such a straightforward sequential training does not guarantee that the model will *remember* all the contributions of previous input data. As shown in the orange curve in Figure 4, the previously learned graph exhibits a larger predictive uncertainty as the training phase proceeds.

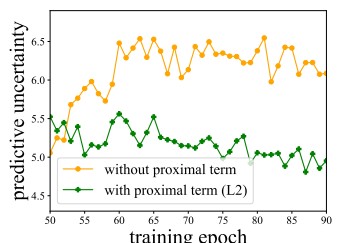

Figure 4: Predictive uncertainty versus training epoch.

The empirical result indicates that the knowledge or information contained in previous input data will be forgotten or covered by newly incoming data. This phenomenon, called *catastrophic forgetting*, was first noticed in continual learning [35] and also

appears in our case. Intuitively, when the training data is taken in a progressive or iterative manner, the learned parameters will cater to the newly incoming data and *forget* the old, previous data points.

One remedy for this issue is adding a proximal term to the objective. The additional proximal term (*i.e.*, the regularization) guarantees the proximity between the new parameters and the model parameters learned from previous graphs. Therefore, when the model is learning the $k$-th input graph, the loss function for our pre-training model in APT is

$$\mathcal{L}(\theta) = \sum_i \mathcal{L}_i(\theta) + \frac{\lambda}{2} \sum_j F_{jj}^{(k-1)}(\theta_j - \theta_j^{(k-1)})^2, \tag{5}$$

where $\mathcal{L}_i$ is given in Eq. (1), the first summation is taken over the subgraph instances sampled from the latest input graph, $\theta^{(k-1)}$ is the model parameters after learning from the first $k-1$ graphs, and the parameter $\lambda$ describes the trade-off between the knowledge learnt from new data and that from previous data. Here, $F^{(k-1)}$ is the Fisher information matrix of $\theta^{(k-1)}$, and $F_{jj}^{(k)}$ is its $j$-th diagonal element. When $F$ is set as an identity matrix, the second term degenerates to the L2 regularization (which serves as one of our variants). The proximal term in Eq. (5) is absent when the first input graph is introduced to the model, and the term is applied on the first three layers of the pre-training model. Finally, we note that the total number of parameters in the pre-training model is in the same order of magnitude as classical GNNs, so the memory cost would not be a bottleneck.

### 3.3 Training and fine-tuning

Integrating the graph selector and the pre-training model forms the entire APT framework, and the overall algorithm is presented in Appendix C. After the training phase, the APT framework returns a pre-trained GNN model, and then the pre-trained model is applied to various downstream tasks from a wide spectrum of domains. In the so-called *freezing mode*, the pre-trained model returned by APT is directly applied to downstream tasks, without any changes in parameters. Alternatively, the *fine-tuning mode* uses the pre-trained graph encoder as initialization, and offers the flexibility of training the graph encoder and the downstream classifier together in an end-to-end manner.

## 4 Experiments

In the experiments, we pre-train a graph representation model via the proposed APT framework, and then evaluate the transferability of our pre-trained model on multiple unseen graphs in the node classification and graph classification task. Lastly, we include the applicable scope of our pre-trained model. Additional experiments can be found in Appendix H, including our adaptation to backbone pre-training models GraphCL, JOAO and Mole-BERT, training time, impact of different graph properties, analysis of the ablation studies and selected pre-training graphs, hyper-parameter sensitivity, explorations of various combinations of graph properties.

### 4.1 Experimental setup

**Datasets.** The datasets for pre-training and testing, along with their statistics, are listed in Appendix D. Pre-training datasets are collected from different domains, including social, citation, and movie networks. Testing datasets comprise both in-domain (*e.g.*, citation, movie) and cross-domain (*e.g.*, image, web, protein, transportation and others) datasets to evaluate transferability comprehensively, also including large-scale datasets with millions of edges sourced from [28].

**Baselines.** We evaluate our model against the following baselines for node and graph classification tasks. For node classification, ProNE [80], DeepWalk [49], struc2vec [53], DGI [65], GAE [34], and GraphSAGE [23] are used as baselines, and then the learned representations are fed into the logistic regression, as most of baselines did. As for graph classification, we take graph2vec [46], InfoGraph [59], DGCNN [81] and GIN [71] as baselines, using SVM as the classifier, which aligns with the methodology of most baselines. For both tasks, we also compare our model with (1) Random: random vectors are generated as representations; (2) GraphCL [76]: a GNN pre-training scheme based on contrastive learning with augmentations; (3) JOAO [75]: a GNN pre-training scheme that automatically selects data augmentations; (4) GCC [50]: the state-of-the-art cross-domain graph pre-training model (which is our model's version without the data selector, trained on all pre-training

Table 1: Micro F1 scores of different models in the node classification task. The column "A.R." reports the average rank of each model. Asterisk (∗) denotes the best result on each dataset, and bold numbers denote the best result among graph pre-training models in the freezing or fine-tuning setting. The notation "/" means out of memory or no convergence for more than three days.

| Dataset / Method | brazil | dd242 | dd68 | dd687 | wisconsin | cornell | cora | pubmed | ogbarxiv | ogbproteins | A.R. |
|---|---|---|---|---|---|---|---|---|---|---|---|
| Random | 32.16(13.65) | 6.71(2.44) | 8.29(4.35) | 5.98(2.70) | 26.79(8.59) | 39.77(7.26) | 26.80(1.62) | 38.85(0.76) | 11.89(4.66) | 52.69(5.94) | 13.2 |
| ProNE | 50.24(11.56) | 10.04(2.56) | 7.73(3.11) | 3.88(1.66) | 44.67(7.49) | 47.32(12.14) | 80.76(2.92)* | 78.80(0.98)* | 65.96(0.06)* | 76.28(0.34)* | 7.5 |
| DeepWalk | 43.16(16.78) | 8.11(1.45) | 6.72(3.04) | 6.17(2.18) | 39.61(9.11) | 47.67(8.30) | 49.85(9.26) | 44.99(10.89) | 16.04(2.98) | 64.74(0.49) | 8.7 |
| struc2vec | 25.54(11.74) | 13.71(2.66) | 10.30(3.23) | 7.98(2.74) | 45.39(7.36) | 38.39(9.18) | 36.01(2.41) | 44.45(0.78) | / | / | 10.6 |
| DGI | 56.44(7.79) | 14.35(0.44) | 13.57(0.44) | 11.04(1.93) | 49.46(5.46) | 49.85(9.26) | 30.02(0.44) | 42.39(0.84) | 12.93(7.67) | 55.98(0.33) | 6.6 |
| GAE | 57.88(10.68) | 14.09(1.52) | 13.43(0.96) | 10.25(2.63) | 45.78(4.18) | 49.26(5.24) | 30.10(0.31) | 40.14(0.68) | / | / | 8.5 |
| GraphSAGE | 67.93(9.28) | 14.33(0.37) | 13.55(0.70) | 10.39(0.78) | 47.03(1.98) | 49.20(1.46) | 35.93(1.76) | 39.94(0.02) | / | / | 7.1 |
| GraphCL (freeze) | 50.71(5.00) | 9.53(2.50) | 9.36(3.63) | 6.03(1.86) | 38.85(10.80) | 41.05(5.67) | 16.95(2.39) | 41.07(1.16) | / | / | 12.2 |
| JOAO (freeze) | 71.22(7.21) | 7.98(2.90) | 12.36(2.59) | 5.34(1.43) | 42.69(8.15) | 43.16(5.67) | 18.13(2.82) | 41.05(0.87) | / | / | 10.9 |
| GCC (freeze) | 67.47(4.09) | 15.83(0.80) | 11.95(1.13) | 9.61(0.94) | 52.57(1.69) | 46.87(1.73) | 35.47(0.51) | 46.40(0.18) | 14.56(7.60) | 59.15(0.35) | 7.0 |
| APT-G (freeze) | 68.69(3.42) | **17.21(1.13)** | 11.98(0.75) | 9.54(1.29) | 54.45(1.90) | 46.53(1.59) | 34.89(0.25) | 46.49(0.22) | 12.32(7.71) | 60.38(0.41) | 6.3 |
| APT-P (freeze) | 66.55(2.35) | 16.58(0.97) | 12.48(0.85) | 10.33(0.83) | 51.90(1.64) | 47.33(2.31) | 35.63(0.56) | 46.16(0.12) | 12.86(7.54) | 60.32(0.32) | 6.2 |
| APT-R (freeze) | 68.12(3.07) | 16.72(0.72) | 12.42(1.24) | **11.05(0.88)** | 54.48(1.77) | 46.80(1.08) | 34.93(0.36) | 46.02(0.11) | 18.79(5.87) | 62.18(0.46) | 5.0 |
| APT-L2 (freeze) | 69.82(2.32) | 16.79(0.88) | **12.68(0.81)** | 10.34(1.12) | **55.11(1.74)** | **48.76(2.20)** | 34.27(0.43) | 46.21(0.15) | 19.64(6.46) | 60.23(0.37) | 4.4 |
| APT (freeze) | **73.39(2.55)** | 16.57(0.94) | 12.08(0.89) | 10.35(1.24) | 53.38(1.19) | 47.37(1.29) | **36.69(0.49)** | **46.88(0.21)** | **22.04(0.29)** | **62.29(0.55)** | 3.8 |
| GraphCL (rand, fine-tune) | 64.43(14.95) | 15.04(0.85) | 14.69(2.48) | 10.99(0.58) | 63.85(2.18) | 44.21(10.58) | 30.45(0.37) | 40.73(0.66) | / | / | 8.3 |
| JOAO (rand, fine-tune) | 72.14(6.74) | 10.93(2.85) | 8.08(2.15) | 7.40(3.48) | 45.38(13.30) | 45.26(10.31) | 29.93(2.84) | 42.01(0.68) | / | / | 9.6 |
| GCC (rand, fine-tune) | 58.51(3.07) | 15.98(1.05) | 13.16(1.06) | 9.74(0.95) | 53.85(2.58) | 50.95(2.26) | 43.70(0.52) | 49.72(0.17) | 18.61(1.88) | 59.12(0.35) | 7.6 |
| GraphCL (fine-tune) | 73.57(10.33) | 15.35(0.99) | 13.51(2.57) | 10.66(1.04) | 63.85(4.42) | 51.05(2.41) | 30.81(0.36) | 42.91(0.91) | / | / | 7.5 |
| JOAO (fine-tune) | 75.00(5.76) | 10.54(3.07) | 7.56(1.94) | 8.77(2.39) | 50.00(12.28) | 42.11(10.26) | 29.34(3.04) | 42.21(0.88) | / | / | 9.5 |
| GCC (fine-tune) | 74.46(3.05) | 19.32(0.80) | 13.87(1.13) | 10.37(1.06) | 59.47(1.49) | 48.32(2.42) | 43.34(0.38) | 50.87(0.19) | 18.62(1.92) | 60.08(0.56) | 6.4 |
| APT-G (fine-tune) | 77.60(1.48) | 25.45(0.60)* | 17.78(1.14) | 11.27(0.76) | 66.09(2.28) | 53.02(1.51) | 45.63(0.66) | 50.81(0.18) | 27.33(4.80) | 60.02(0.32) | 3.8 |
| APT-P (fine-tune) | 78.99(2.44) | 25.19(0.87) | 16.40(1.22) | 11.69(1.19) | 64.24(1.90) | 50.05(1.39) | 45.53(0.30) | 50.66(0.18) | 27.20(4.80) | 59.86(0.32) | 4.8 |
| APT-R (fine-tune) | 79.14(1.97) | 24.96(0.57) | 17.43(1.05) | 11.29(1.04) | 66.28(1.94) | **53.56(2.28)*** | 46.02(0.83) | 51.00(0.21) | 18.41(1.84) | 60.10(0.38) | 3.4 |
| APT-L2 (fine-tune) | 78.75(1.63) | 24.62(0.90) | 17.83(1.35)* | 12.26(0.78)* | 67.04(1.50)* | 52.94(1.95) | 47.48(0.46) | 51.25(0.21) | 27.40(4.97) | 60.85(0.46) | 2.6 |
| APT (fine-tune) | **79.67(2.30)*** | **28.62(0.55)*** | **20.30(1.13)*** | **12.80(1.54)*** | **67.08(1.75)*** | 52.15(2.25) | **47.51(0.62)** | **51.30(0.16)** | **27.40(4.87)** | **61.64(0.35)** | **1.3** |

data). GCC, GraphCL and JOAO are trained on the entire collected input data, and the suffix (rand, fine-tune) indicates whether the model is trained from scratch. We also include 9 variants of our model: (1) APT-G: removes the criteria of graph properties in the graph selector; (7) APT-P: removes the criteria of predictive uncertainty in the graph selector; (8) APT-R: removes the regularization w.r.t old knowledge in Eq. (5); (8) APT-L2: degenerates the second term in Eq. (5) with L2 regularization.

**Experimental settings.** In the training phase, we aim to utilize data from different domains to pre-train one graph model. We iteratively select graphs for pre-training until the predictive uncertainty of any candidate graph is below 3.5. For each selected graph, we choose samples with predictive uncertainty higher than 3. We set the number of subgraph instances queried in the graph for uncertainty estimation $M$ as 500. The time-adaptive parameter $\gamma_t$ in Eq. (4) follows a $\gamma_t \sim \text{Beta}(1, \beta_t)$, where $\beta_t = 3 - 0.995^t$. We set the trade-off parameter $\lambda = 10$ for APT-L2, and $\lambda = 500$ for APT. The total iteration number is 100. We adopt GCC as the backbone pre-training model with its default hyper-parameters, including their subgraph instance definition. In the fine-tuning phase, we select logistic regression or SVM as the downstream classifier and adopt the same setting as GCC. Due to space limit, results of our adaption on other backbones like GraphCL, JOAO and Mole-BERT, as well as details can be found in Appendices H and G. See Appendix G for details.

We provide an open-source implementation of our model APT at https://github.com/galina0217/APT.

## 4.2 Experimental results

**Node classification.** Table 1 presents the micro F1 score of different methods over 10 unseen graphs from a wide spectrum of domain for node classification task. We observe that our model beats the graph pre-training competitor by an average of 9.94% and 17.83% under freezing and fine-tuning mode respectively. This suggests that instead of pre-training on all the collected graphs (like GCC), it is better to choose a part of graphs better suited for pre-training (like our model APT).

Moreover, compared with the traditional models without pre-training, the performance gain of our model is attributed to the transferable knowledge learned by pre-training strategies. We also find that some proximity-based models, like ProNE, enforce neighboring nodes to share similar representations, leading to superior performance on graphs with strong homophily rather than weak homophily. Nonetheless, when applied to different graphs, these models require re-training, which makes them non-transferable. In contrast, we target at a general, transferable model free from specific assumptions on graph types, and thus applicable across wide scenarios (including both homophilic and heterophilic graphs). Ablation study reveals the necessity of all the components in APT (*e.g.,*

the proximity regularization with respect to old knowledge, proximity term, graph properties and predictive uncertainty). We also explore the impacts of the five graph properties used in our model, and demonstrate their indispensability by experiments. Thus, combining all graph properties is essential to boost the performance of APT. Details in ablation study and more experimental results can be found in Appendix H.

**Graph classification.** The micro F1 score on unseen test data in the graph classification task is summarized in Table 2. Especially, our model is 7.2% and 1.3% on average better than the graph pre-training backbone model under freezing and fine-tuning mode, respectively. Interestingly, we observe that all variants of APT perform quite well, and thus, a simpler yet well-performed version of APT could be used in practice. This phenomenon could happen since "graph pre-train and fine-tune" is an extremely complicated non-convex optimization problem. Another observation is that in specific cases, such as dd dataset, a decrease in downstream performance after fine-tuning is observed, as compared to the freezing mode. This could happen since "graph pre-train and fine-tune" is an extremely complicated non-convex optimization problem. A similar observation has been made in previous work [37] as well.

Finally, it is worth noting that our APT model achieves a training time 2.2× faster than the competitive model GCC, achieved through a reduced number of carefully selected training graphs and samples. More specifically, we carefully selected only 7 datasets out of the available 11 and performed pre-training using at most 24.92% of the samples in each selected dataset. Moreover, for each newly added dataset, our model only needs a few more training iterations to convergence, rather than being trained from scratch.

### 4.3 Discussion: scope of application

The transferability of the pre-trained model comes from the learned representative structural patterns and the ability to distinguish these patterns (as discussed in §2). Therefore, our pre-training model is more suitable for the datasets where the target (*e.g.*, labels) is correlated with

Table 2: Micro F1 of different models in graph classification.

| Dataset / Method | imdb-binary | dd | msrc-21 | A.R. |
|---|---|---|---|---|
| Random | 49.30(4.82) | 52.72(4.34) | 4.49(2.14) | 11 |
| graph2vec | 56.20(5.33) | 59.16(3.47) | 8.22(3.67) | 7.7 |
| InfoGraph | 66.58(0.63) | 58.66(0.23) | 6.01(0.59) | 7.7 |
| GraphCL (freeze) | 55.10(3.18) | 57.82(4.71) | 5.44(2.77) | 9.3 |
| JOAO (freeze) | 63.90(3.48) | 55.97(3.61) | 5.09(2.65) | 9.3 |
| GCC (freeze) | 73.09(0.55) | 75.16(0.53) | 11.61(1.33) | 5.3 |
| APT-G (freeze) | 73.10(0.39) | 75.24(0.42) | 12.81(0.74) | 4.3 |
| APT-P (freeze) | 72.83(0.81) | **76.38(0.32)** | 13.30(0.57) | 3.0 |
| APT-R (freeze) | **73.98(0.21)** | 75.32(0.34) | 12.90(0.57) | 3.0 |
| APT-L2 (freeze) | 73.54(0.40) | 75.81(0.30) | 13.16(0.77) | 2.3 |
| APT (freeze) | 73.00(0.50) | 75.83(0.31) | **13.81(1.06)** | 3.0 |
| DGCNN | 71.00(4.69) | 58.63(4.46) | 6.01(0.59) | 11.3 |
| GIN | 72.00(2.41) | 77.61(1.47)* | 10.54(5.08) | 6.0 |
| GraphCL (rand, fine-tune) | 63.60(3.61) | 58.15(4.60) | 8.25(2.94) | 12.7 |
| JOAO (rand, fine-tune) | 67.70(3.35) | 62.10(4.31) | 11.40(3.06) | 10.0 |
| GCC (rand, fine-tune) | 75.80(1.37) | 74.26(0.59) | 17.18(1.43) | 7.3 |
| GraphCL (fine-tune) | 66.90(4.39) | 65.55(5.14) | 8.77(2.60) | 10.7 |
| JOAO (fine-tune) | 68.50(3.61) | 62.61(4.99) | 10.18(1.72) | 10.0 |
| GCC (fine-tune) | 76.19(0.90) | 75.32(1.77) | 24.90(1.65) | 4.7 |
| APT-G (fine-tune) | 76.29(0.89) | 75.46(0.77) | 21.94(0.73) | 4.7 |
| APT-P (fine-tune) | **76.70(1.01)*** | 75.34(0.88) | 24.32(1.22) | 3.7 |
| APT-R (fine-tune) | 76.60(1.02) | 75.64(0.70) | 24.09(2.12) | 3.3 |
| APT-L2 (fine-tune) | 75.93(0.84) | 75.58(1.06) | **25.58(1.57)*** | 3.7 |
| APT (fine-tune) | 76.27(1.20) | **75.69(1.42)** | 24.41(1.82) | 3.0 |

subgraph patterns or structural properties (*e.g.*, motifs, triangles, betweenness, stars). For example, for node classification on heterophilous graphs (*e.g.*, winconsin, cornell), our model performs very well because in these graphs, nodes with the same label are not directly connected, but share similar structural properties and behavior (or role, position). On the contrary, graphs with strong homophily (like cora, pubmed, ogbarxiv and ogbproteins) may not benefit too much from our models. Similar observation can also be made on graph classification: our model could also benefit the graphs whose label has a strong relationship with their structure, like molecular, chemical, and protein networks (*e.g.*, dd in our experiments) [17, 66].

## 5 Related Work

**Graph pre-training.** Inspired by pre-training in CV/NLP, recent efforts have shed light on pre-training on GNNs. Initially, unsupervised graph representation learning is used for graph pre-training [13, 21, 23, 46, 53, 63, 80]. The design of unsupervised models is largely based on the neighborhood similarity assumption, and thus cannot generalize to unseen graphs. More recently, self-supervised graph learning emerges as another line of research, including graph generative and contrastive models. Graph generative models aim to capture the universal graph patterns by recovering certain parts of input graphs [7, 27, 30, 34, 67], but they rely heavily on domain-specific knowledge. In comparison, contrastive models maximize the agreement between positive pairs and minimize that between negative pairs [24, 25, 29, 39, 40, 44, 59, 60, 65, 69, 70, 76, 78, 83, 85–87]. Some work in

this direction takes subgraph sampling as augmentation, in the hope that the transferable subgraph patterns can be captured during pre-training. However, all the aforementioned studies only focus on the design of pre-training models, rather than suitable selection of data for pre-training.

**Uncertainty-based sample selection.** The terminology *uncertainty* is widely used in machine learning, without a universally-accepted definition. In general, this term refers to the lack of confidence of an ML model in certain model parameters. The majority of existing works define uncertainty in the label space, such as taking the uncertainty as the confidence level about the prediction [16, 18, 42, 61]. Only a few works define uncertainty in the representation space [52, 68]. In [52], uncertainty is measured based on the representations of an instance's nearest neighbors with the same label. However, this approach requires access to the label information of the neighbors, and thus cannot be adapted in pre-training with unlabeled data. [68] introduces a pretext task for training a model of uncertainty over the learned representations, but this method assumes a well-pre-trained model is already available. Such a post processing manner is not applicable to our scenario, because we need an uncertainty that can guide the selection of data during pre-training rather than after pre-training.

Some works on active learning and hard example mining (HSM) have also introduced concept similar to uncertainty. In active learning, uncertainty is measured via classification prediction, and the active learning model focuses on those samples which the model least certain about [4, 74, 82, 84]. However, these techniques all rely on the labels and cannot be adapted in pre-training with unlabeled data. As another line of work, HSM introduces similar strategies and works on those samples with the greatest loss, which can also be regarded as a kind of uncertainty [36, 43, 55, 56]. Nevertheless, existing HSM approaches do not meet the following two requirements needed in our setting. (1) The chosen instances should follow a joint distribution that reflects the topological structures of real-world graphs. This is satisfied by our use of graph-level predictive uncertainty and graph properties, but is not met in HSM. (2) The chosen set of graphs should include informative and sufficiently diverse instances. This is only achieved by the proposed APT framework while existing HSM methods fail to consider this requirement.

**Pre-training in CV and NLP.** For pre-training in CV and NLP, scaling up the pre-training data size often results in a better or saturating performance in the downstream [1, 15, 33, 51, 62]. In view of this, data selection is not an active research direction for CV and NLP. Existing studies mainly focus on selecting pre-training data that closely matches the downstream domain [2, 5, 8–10, 38, 72]. The assumption on downstream domain knowledge differs from our graph pre-training setting, making such data selection less relevant to our work.

**Data-centric AI.** This recently introduced concept emphasizes the enhancement of data quality and quantity, rather than model design [73, 79]. Following-up works in graph pre-training [24, 64] exploits the data-centric idea to design data augmentation. For example, [24] introduces a graph data augmentation method by interpolating the generator of different classes of graphs. [64] mainly focuses on the theoretical analysis of data-centric properties of data augmentation. While many of these works advocate for shifting the focus to data, they do not consider the co-evolution of data and model, as is the case in our work.

## 6 Conclusion

In this paper, we identify the *curse of big data* phenomenon for pre-training graph neural networks (GNNs). This observation then motivates us to choose a few suitable graphs and samples for GNN pre-training rather than training on a massive amount of unselected data. Without any knowledge of the downstream tasks, we propose a novel graph selector to provide the most instructive data for pre-training. The pre-training model is then encouraged to learn from the data in a progressive and iterative way, reinforce itself on newly selected data, and provide instructive feedback to the graph selector for further data selection. The integration of the graph selector and the pre-training model into a unified framework forms a data-active graph pre-training (APT) paradigm, in which the two components are able to mutually boost the capability of each other. Extensive experimental results verify that the proposed APT framework indeed enhances model capability with fewer input data.

**Acknowledgments**

This work was partially supported by NSFC (62206056, 62322606). Xin Jiang and Carl Yang were not supported by any fund from China.

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

## A  Theoretical Connection Between Network Entropy and Typical Graph Properties

Many interesting graph structural properties from basic graph theory give rise to a graph with high network entropy [19]. We here theoretically show some connections between the proposed network entropy and typical structural properties.

To make theoretical analysis, we consider connected, unweighted and undirected graph, whose network entropy depends solely on its degree distribution (see Eq. (3)). Considering a random graph $G$ with a fixed node set, we suppose that the degree of any node $v_i$ independently follows distribution $p$, which is a common setting in random graph theory [8]. Then the expected network entropy of $G$ is

$$\langle \mathcal{H}(G) \rangle = \frac{1}{2|E|} \sum_i \langle \boldsymbol{d}_i \log \boldsymbol{d}_i \rangle = \frac{\langle \boldsymbol{d} \log \boldsymbol{d} \rangle}{\langle \boldsymbol{d} \rangle}. \tag{6}$$

where every $\boldsymbol{d}_i$ (and $\boldsymbol{d}$) is an independent random variable follows the distribution $p$.

Now we are ready to discuss the connection between network entropy $\langle \mathcal{H}(G) \rangle$ and some typical graph properties (*i.e.*, average degree $\langle \boldsymbol{d} \rangle$, degree variance $\mathrm{Var}(\boldsymbol{d})$ and the scale-free exponent $\alpha$).

**Average degree.**  Given that the function $x \log x$ is convex in $x$, we have

$$\langle \mathcal{H}(G) \rangle \geq \frac{\langle \boldsymbol{d} \rangle \log \langle \boldsymbol{d} \rangle}{\langle \boldsymbol{d} \rangle} = \log \langle \boldsymbol{d} \rangle. \tag{7}$$

It is clear that average degree is the lower bound of network entropy. Based on our discussion on §3.1, we conclude that when used for pre-training, an input graph with higher average degree would in general result in better performance of the pre-trained model.

**Degree variance.**  The Taylor expansion of $\langle \boldsymbol{d} \log \boldsymbol{d} \rangle$ in Eq. (6) at $\langle \boldsymbol{d} \rangle$ gives

$$\langle \mathcal{H}(G) \rangle = \log \langle \boldsymbol{d} \rangle + \frac{\mathrm{Var}(\boldsymbol{d})}{2 \langle \boldsymbol{d} \rangle^2} + o \left( \frac{1}{\langle \boldsymbol{d} \rangle^2} \right).$$

where $\mathrm{Var}(\boldsymbol{d})$ is the variance of $\boldsymbol{d}$. We find that $\log \langle \boldsymbol{d} \rangle$ is exactly the zeroth-order term in the expansion. When average degree is fixed, the network entropy and the degree variance $\mathrm{Var}(\boldsymbol{d})$ are positively correlated. This in turn implies a positive correlation between degree variance and the test performance of the model.

**Scale-free exponent.**  Most real-world networks exhibit an interesting *scale-free* property (*i.e.*, only a few nodes have high degrees), and thus the degree distribution often follows a power-law distribution. That is, we can just write the degree distribution as $p(x) \sim x^{-\alpha}$, where $\alpha$ is called the *scale-free exponent*. For a real-world network, the scale-free exponent $\alpha$ is usually larger than 2 [4]. Suppose the degrees of a random graph $G$ with $N$ nodes follows a power-law distribution $p(x) = Cx^{-\alpha}$ where $C$ is a normalization constant. When $\alpha > 2$, we could approximately have [8]

$$\langle \mathcal{H}(G) \rangle = \frac{1}{\alpha - 2}, \quad \text{if } N \to \infty.$$

Clearly, a smaller scale-free exponent $\alpha$ results in a higher network entropy.

**Remark 1** (Connection between network entropy and typical structural properties)*. A graph with high network entropy arises from graphs with typical structural characteristics like large average degree, large degree variance, and scale-free networks with low scale-free exponent.*

Besides the above theoretical analysis. The motivation of choosing density, average degree, degree variance and scale-free exponent is similar to that of network entropy. Intuitively, graphs with larger *average degree* and higher *density* have more interactions among the nodes, thus providing more topological information to graph pre-training. Also, the larger the diversity of node degrees, the more diverse the subgraph samples. The diversity of node degrees can be measured by *degree variance* and *scale-free exponent*. (A smaller scale-free exponent indicates the length of the tail of degree distribution is relatively longer, *i.e.*, the degree distribution spreads out wider.)

# B  Theoretical Analysis and Advantages of Predictive Uncertainty

In this section, we provide the proofs of Theorem 1 and demonstrate the advantages of our proposed uncertainty.

**Proofs of Theorem 1**  Before we start the proof for Theorem 1, we introduce three useful lemmas that are adopted to prove the theorem as follows.

**Lemma 1** (Fano's Inequality.). *Let $X$ be a random variable uniformly distributed over a finite set of outcomes $\mathcal{X}$. For any estimator $\hat{X}$ such that $X \to Y \to \hat{X}$ forms a Markov chain, we have*

$$\Pr(\hat{X} \neq X) \geq 1 - \frac{\mathrm{I}(X;\hat{X}) - \log 2}{\log |\mathcal{X}|}. \tag{8}$$

**Lemma 2** (Data-Processing Inequality.). *For any Markov chain $X \to Y \to Z$, we have*

$$\mathrm{I}(X;Y) \geq I(X;Z) \text{ and } \mathrm{I}(Y;Z) \geq \mathrm{I}(X;Z). \tag{9}$$

The proof of lemma 1 and lemma 2 can be found under Chapter 2 in [5]. The above lemmas establish a connection between mutual information and the accuracy of downstream task. Next, we further establish the connection between mutual information and ours uncertainty measure.

**Lemma 3** (Connection between InfoNCE and mutual information). *Assume that graph encoder $g$ is a scaling function, namely $g(\boldsymbol{x}) = k\boldsymbol{x}$. Hence, the mutual information $\mathrm{I}(X;Z)$ can be expressed by the InfoNCE loss $\phi_{\mathrm{uncertain}}$ for node in Eq.(1) as follows:*

$$-\phi_{\mathrm{uncertain}} \geq -\mathrm{I}(X;f(X)) = -\mathrm{I}(X;Z). \tag{10}$$

***Proof:*** *We follow [20] to prove the theorem. The mutual information $\mathrm{I}(X;Z)$ can be expressed as:*

$$-\phi_{\mathrm{uncertain}} := \sup_f \mathbb{E}_{(x_i,g(x_i))\sim P_{X,g(X)}} \left[ \frac{1}{n} \sum_{i=1}^n \log \frac{e^{f(x_i,g(x_i))}}{\sum_{j=1}^n e^{f(x_i,g(x_j))}} \right] \tag{11}$$
$$\leq D_{\mathrm{KL}}\left(P_{X,g(X)} \| P_X P_{g(X)}\right) = \mathrm{I}(X;g(X)) = \mathrm{I}(X;Z),$$

*where $f(x;y)$ is any similarity scoring function like $f(x;y) = \cos(x)\cos(y)/\tau$.*

Therefore, we can now prove that lower existing uncertainty $\mathcal{L}_{\mathrm{CE}}(\zeta)$ over all downstream classifiers cannot be achieved without lower uncertainty $\phi_{\mathrm{uncertain}}(\zeta)$.

**Theorem 1** (Theoretical connection between uncertainty.). *Let $\mathcal{X}$, $\mathcal{Z}$ and $\mathcal{Y}$ be the input space, representation space and label set of downstream classifier. Assume that the distribution of labels is a uniform distribution over $\mathcal{Y}$. Consider the set of downstream classifiers $\mathcal{H} = \{h : \mathcal{Z} \to \mathcal{Y}\}$. For any graph encoder $f : \mathcal{X} \to \mathcal{Z}$,*

$$\mathcal{L}_{\mathrm{CE}}(\zeta) \geq \log\left(\frac{\log |\mathcal{Y}|}{\log 2 - \phi_{\mathrm{uncertain}}(\zeta)}\right), \tag{12}$$

*where $\mathcal{L}_{\mathrm{CE}}$ denotes the conventional uncertainty (which is the cross entropy loss here) estimated from the composition of graph encoder and downstream classifier $h \circ f$, and $\phi_{\mathrm{uncertain}}$ is our proposed uncertainty estimated from graph encoder $f$.*

*Proof.* For $h \in \mathcal{H}$, we have the Markov chain

$$Y \to X \xrightarrow{f} f(X) \xrightarrow{h} (h \circ f)(X),$$

where $X, Y$ are random variables for input and label distributions respectively. The first Markov chain $Y \to X$ can be understood as a generative model for generating inputs according to the conditional probability distribution $\mu_{X|Y}$. Therefore, applying Lemmas 1 and 2, we obtain the inequality,

$$\Pr\left[h\left(f\left(X\right)\right) \neq Y\right] \geq 1 - \frac{\mathrm{I}\left(Y;(h \circ f)(X)\right) + \log 2}{\log |\mathcal{Y}|} \geq 1 - \frac{\mathrm{I}(X;f(X)) + \log 2}{\log |\mathcal{Y}|}. \tag{13}$$

$\Pr\left[h\left(f\left(X\right)\right) \neq Y\right]$ represents the error rate. According to the definition of cross entropy, the cross entropy for node can be denoted as follows:

$$\mathcal{L}_{\mathrm{CE}} = -\log(1 - \Pr\left[h\left(f\left(X\right)\right) \neq Y\right]). \tag{14}$$

By combining Eq. (13), Eq. (10) with Eq. (14), we have

$$\Pr\left[h\left(f\left(X\right)\right) \neq Y\right] = 1 - e^{-\mathcal{L}_{\mathrm{CE}}(\zeta)} \geq 1 - \frac{\mathrm{I}(X; Z) + \log 2}{\log |\mathcal{Y}|} \geq 1 + \frac{\phi_{\mathrm{uncertain}}(\zeta) - \log 2}{\log |\mathcal{Y}|},$$

$$\mathcal{L}_{\mathrm{CE}}(\zeta) \geq \log(\frac{\log |\mathcal{Y}|}{\log 2 - \phi_{\mathrm{uncertain}}(\zeta)}). \tag{15}$$

Thus, we completed the proof. □

**Advantages of our proposed uncertainty** Except for the theoretical advantages, we further discuss two advantages of using the model loss (*i.e.*, InfoNCE loss) as predictive uncertainty for data selection. First, InfoNCE loss is exactly the objective function of our model, so what we do is actually to select the samples with the greatest contributions to the objective function. Such strategy has been justified to accelerate convergence and enhance the discriminative power of the learned representations [18, 28, 29, 31]. Second, as the loss function of our model, InfoNCE is already computed during the training, and thus no additional computation expense is needed in the data selection phase.

## C Algorithm

The overall algorithm for APT is given in Algorithm 1. Given a collection of graphs $\mathcal{G} = \{G_1, \ldots, G_N\}$ from various domains, APT aims to pre-train a better generalist GNN (*i.e.*, pre-training model) on wisely chosen graphs and samples. Our APT pipeline involves the following three steps. (i) At the beginning, the graph selector chooses a graph for pre-training according to the graph properties (line 1). (ii) Given the chosen graph, the graph selector chooses the subgraph samples whose predictive uncertainty is higher than $T_s$ in this graph (line 3). (iii) The selected samples are then fed into the model for pre-training until the predictive uncertainty of the chosen graph is below $T_g$ or the number of training iterations on this chosen graph reaches $F$ (line 4-5). (iv) The model's feedback in turn helps select the most needed graph based on predictive uncertainty and graph properties until the predictive uncertainty of any candidate graph is low enough (line 6-7). The last three steps are repeated until the iteration number reaches a pre-set maximum value T (which can be considered as the total iteration number required to train on all selected graphs).

---

**Algorithm 1** Overall algorithm for APT.

---

**Input:** A collection of graphs $\mathcal{G} = \{G_1, \ldots, G_N\}$, maximal period $F$ of training one graph, trade-off parameter $\gamma_t = 0$, hyperparameter $\{\beta_t\}$, the learning rate $\mu$, the predictive uncertainty threshold of moving to a new graph $T_g$, the predictive uncertainty threshold of choosing training samples $T_s$, and the maximum iteration number $T$.
**Output:** Model parameter $\theta$ of the pre-trained graph model.

1: Choose a graph $G^*$ from $\mathcal{G}$ via the graph selector, and $\mathcal{G} \leftarrow \mathcal{G} \backslash \{G^*\}$.
2: **while** The iteration number reaches $T$ **do**
3:      Sample instances with predictive uncertainty higher than $T_s$ from $G^*$ via the graph selector.
4:      Update model parameters $\theta \leftarrow \theta - \mu \nabla_\theta \mathcal{L}(\theta)$.
5:      **if** $\phi_{\mathrm{uncertain}}(G^*) < T_g$ *or* the model has been trained on $G^*$ by $F$ iterations **then**
6:          Update the trade-off parameter $\gamma_t \sim \mathrm{Beta}\left(1, \beta_t\right)$.
7:          Choose a graph $G^*$ from $\mathcal{G}$, and $\mathcal{G} \leftarrow \mathcal{G} \backslash \{G^*\}$.
8:      **end if**
9: **end while**

---

The time complexity of our model mainly consists of five components: data augmentation, GNN encoder propagation, contrastive loss, sample selection and graph selection. Suppose the maximal number of nodes of subgraph instances is $|V|$, the batch size is $B$, and $D$ is the representation dimension. (1) As for the data augmentation, the time complexity of random walk with restart is at least $O(B|V|^3)$ [34]. (2) The time complexity of GNN encoder propagation depends on the architectures of the backbone GNN. We denote it as $X$ here. (3) The time complexity of the contrastive loss is $O(B^2 D)$ [17]. (4) Sample selection is conducted by choosing the samples with high contrastive loss (the loss is computed before), which costs $O(B)$. (5) Graph selection costs

Table 3: Datasets for pre-training and testing, where * denotes the average statistic of multiple graphs under graph classification setting. $|V|$ and $|E|$ denote the number of nodes and the number of edges in a graph, respectively.

|  | Type | Name | $|V|$ | $|E|$ | Description |
|---|---|---|---|---|---|
| **pre-training data** | citations | arxiv | 86,376 | 517,563 | citations between papers on the arxiv |
|  |  | dblp | 93,156 | 178,145 | same as above (dblp) |
|  |  | patents-main | 240,547 | 560,943 | citations between US patents |
|  | social | soc-sign0902 | 81,867 | 497,672 | friend/foe links between the users of Slashdot in Feb. 2009 |
|  |  | soc-sign0811 | 77,350 | 468,554 | same as above (Nov. 2008) |
|  |  | wiki-vote | 7,115 | 100,762 | voting relationships between wikipedia users |
|  |  | academia | 137,969 | 369,692 | friendships between academics on Academia.edu |
|  |  | michigan | 30,147 | 1,176,516 | friendships between Facebook users in University of Michigan |
|  |  | msu | 32,375 | 1,118,774 | same as above (Michigan State University) |
|  |  | uillinois | 30,809 | 1,264,428 | same as above (University of Uillinois) |
|  | movie | imdb | 896,305 | 3,782,447 | relationships between actors and movies |
| **test data** | protein | dd | 284.32* | 715.66* | molecular interactions between amino acids |
|  |  | ogbproteins | 132,534 | 39,561,252 | biologically associations between proteins |
|  | image | msrc-21 | 77.52* | 198.32* | adjacency between superpixels of the image segmentations |
|  | movie | imdb-binary | 19.77* | 96.53* | collaboration relationships between actors/actresses |
|  | citations | cora | 2,708 | 5,278 | citations between Machine Learning papers |
|  |  | pubmed | 19,717 | 88,648 | citations between scientific papers |
|  |  | ogbarxiv | 169,343 | 1,166,243 | citation network between computer science arxiv papers |
|  | web | cornell | 183 | 280 | hyperlinks between webpages collected from Cornell University |
|  |  | wisconsin | 251 | 466 | same as above (Wisconsin University) |
|  | transportation | brazil | 131 | 2,077 | commercial flights between airports in Brazil |
|  | others | dd242 | 1,284 | 3,303 | this network dataset is in the category of labeled networks |
|  |  | dd68 | 775 | 2,093 | same as above |
|  |  | dd687 | 725 | 2,600 | same as above |

$O(|\mathcal{G}|M^2 D)$ (where $M$ the number of samples needed to compute the predictive uncertainty of a graph, and $|\mathcal{G}|$ is the number of graphs that have not been selected). This step is executed only in a few epochs (around 6% in our current model), so we ignore its time overhead in graph selection. Therefore, the overall time complexity of APT in each batch is $O(B|V|^3 + X + B^2 D + B)$.

## D   Dataset Details

The graph datasets for pre-training and testing in this paper are collected from a wide spectrum of domains (see Table 3 for an overview). The consideration of the graphs for pre-training and test is as follows. When selecting pre-training data, we hope that the graph size is at least hundreds of thousands to contain enough information for pre-training. When selecting test data, we hope that: (1) some test data is in the same domain as the pre-training data, and some is cross-domain, so as to comprehensively evaluate our model's in and across-domain transferability. Accordingly, the in-domain test data is selected from the type of movie and citations, and the others test data are across-domain; (2) the size of test graphs can scale from hundreds to millions, including large-scale datasets with millions of edges from Open Graph Benchmark [10].

Regarding the pre-training datasets, arxiv, dblp and patents-main are citation networks collected from [2], [37] and [9], respectively. Imdb is the collection of movie from [27]. As for the social networks, soc-sign0902 and soc-sign0811 are collected from [16], wiki-vote is from [15], academia is from [7], and michigan, msu and uillions are from [32]. Regarding the test datasets, we collect the protein network dd and ogbproteins from [6] and [10]. The image network msrc-21 is from [23]. The movie network imdb-binary is from [36]. The citation networks, cora, pubmed and ogbarxiv, are from [21], [22] and [10]. The web networks cornell and wisconsin are collected from [24]. The transportation network brazil is form [26], and dd242, dd68 and dd687 are from [27].

The detailed graph properties of the pre-training data and test data are presented in Table 4 and Table 5, respectively.

Table 4: Detailed structural properties of pre-training datasets, where avg properties equals to $\text{MEAN}(\hat{\phi}_{\text{entropy}}, \hat{\phi}_{\text{density}}, \hat{\phi}_{\text{avg\_deg}}, \hat{\phi}_{\text{deg\_var}}, -\hat{\phi}_{\alpha})$ in Eq. (4), and $nei_2$ denotes the average number and standard deviation of $2-$hop neighbors, $|V|$ and $|E|$ denote the number of nodes and the number of edges in a graph, respectively.

| Properties
Dataset | $|V|$ | $|E|$ | avg properties | avg degree | degree var | density | entropy | $\alpha$ | $nei_2$ (avg, std) | avg clustering coef |
|---|---|---|---|---|---|---|---|---|---|---|
| soc-sign0902 | 81867 | 497672 | **-0.32** | 13.16 | 1643.20 | 1.49e-04 | 3.91 | 1.51 | 1192.33, 2305.99 | 0.06 |
| soc-sign0811 | 77350 | 468554 | **-0.32** | 13.12 | 1631.77 | 1.57e-04 | 3.93 | 1.52 | 1226.93, 2312.03 | 0.05 |
| imdb | 896305 | 3782447 | **-0.66** | 9.44 | 298.27 | 9.42e-06 | 3.07 | 1.53 | 316.16, 614.03 | 5e-05 |
| patent | 240547 | 560943 | **-0.96** | 5.66 | 34.95 | 1.94e-05 | 2.04 | 1.57 | 117.23, 172.40 | 0.08 |
| academia | 137969 | 369692 | **-0.89** | 6.36 | 102.14 | 3.88e-05 | 2.38 | 1.57 | 101.40, 225.96 | 0.14 |
| wiki-Vote | 7115 | 103689 | **0.74** | 29.32 | 3314.79 | 4.10e-03 | 4.46 | 1.40 | 972.03, 1045.13 | 0.14 |
| dblp | 93156 | 178145 | **-1.12** | 4.82 | 58.05 | 4.11e-05 | 2.16 | 1.72 | 58.85, 90.46 | 0.27 |
| arxiv | 86376 | 517563 | **-0.43** | 12.98 | 382.12 | 1.39e-04 | 3.22 | 1.41 | 145.21, 309.12 | 0.68 |
| michigan | 30147 | 1176516 | **1.40** | 79.05 | 6369.17 | 2.59e-03 | 4.78 | 1.23 | 4683.90, 3655.25 | 0.21 |
| msu | 32375 | 1118774 | **1.13** | 70.11 | 5087.53 | 2.13e-03 | 4.62 | 1.23 | 4567.63, 3465.16 | 0.21 |
| uillinois | 30809 | 1264428 | **1.44** | 83.08 | 6306.02 | 2.66e-03 | 4.78 | 1.22 | 5267.10, 3831.37 | 0.21 |

Table 5: Detailed structural properties of test datasets, where $nei_2$ denotes the average number and standard deviation of $2-$hop neighbors, and the numbers with $*$ denote the average statistics of multiple graphs under graph classification setting. $|V|$, $|E|$ and $|G|$ denote he number of nodes in a graph, the number of edges in a graph and the number of graphs in graph classification datasets, respectively.

| Properties
Dataset | $|V|$ | $|E|$ | $|G|$ | avg degree | degree var | density | entropy | $nei_2$ (avg, std) | avg clustering coef | # of classes |
|---|---|---|---|---|---|---|---|---|---|---|
| imdb-binary | 19.77* | 193.06* | 1000 | 9.89* | 116.01* | 1.04* | 1.07* | 24.89*,15.91* | 0.95* | 2 |
| msrc-21 | 77.52* | 198.32* | 563 | 6.10* | 30.26* | 6.81e-02* | 1.71* | 17.00*, 5.81* | 0.51* | 20 |
| dd | 284.32* | 715.66* | 1178 | 6.00* | 27.60* | 2.78e-02* | 1.65* | 14.30*,5.68* | 0.48* | 2 |
| cora | 2708 | 5278 | / | 4.90 | 42.53 | 1.44e-03 | 1.71 | 34.98,47.70 | 0.24 | 7 |
| pubmed | 19717 | 44327 | / | 5.50 | 75.44 | 2.28e-04 | 2.23 | 57.10,82.72 | 0.06 | 3 |
| brazil | 131 | 1074 | / | 16.85 | 539.18 | 1.26e-01 | 3.14 | 92.27,28.50 | 0.66 | 4 |
| dd242 | 1284 | 3303 | / | 6.14 | 28.80 | 4.01e-03 | 1.68 | 14.57,4.30 | 0.47 | 20 |
| dd68 | 775 | 2093 | / | 6.40 | 33.42 | 6.98e-03 | 1.76 | 17.40,8.93 | 0.44 | 20 |
| dd687 | 725 | 2600 | / | 8.17 | 55.78 | 9.91e-03 | 2.01 | 25.45,9.96 | 0.48 | 20 |
| wiscosin | 251 | 466 | / | 4.65 | 76.26 | 1.49e-02 | 1.84 | 68.04,58.22 | 0.23 | 5 |
| cornell | 183 | 280 | / | 4.04 | 58.48 | 1.68e-02 | 1.74 | 54.09,44.30 | 0.18 | 5 |
| ogbarxiv | 16343 | 1157799 | / | 14.67 | 4898.17 | 8.07e-05 | 3.63 | 3483.08,6711.40 | 0.23 | 40 |
| ogbproteins | 132534 | 39561252 | / | 598.00 | 742637.58 | 4.50e-03 | 6.84 | 32265.17,19401.46 | 0.28 | 2 |

# E  Additional Observations of *Curse of Big Data* Phenomenon

This section provides more comprehensive observations to support the *curse of big data* phenomenon in graph pre-training, *i.e.*, more training samples and graph datasets do not necessarily lead to better downstream performance.

We investigate 3630 experiments with GCC [25] and GraphCL [39] model with different model configurations (*i.e.*, the number of GNN layers is set to be 3, 4 and 5 respectively), when pre-trained on all training graphs listed in Table 3 and evaluated on different test graphs (annotated in the upper left corner of each figure) under freezing setting. For each experiment, we calculate the mean and standard deviation over 10 evaluation results of the downstream task with random training/testing splits.

The observations of GCC and GraphCL model can be found in Figure 6 and Figure 7 respectively. The downstream results of different test data are presented in separate rows. The figures in left three columns present the effect of scaling up the number of graphs on the downstream performance under different model configurations (*i.e.*, the number of GNN layers) respectively. We first pre-train the model with only two input graphs, and the result is plotted in a dotted line. The largest standard deviation among the results w.r.t different graph last is also marked by the blue arrow. The figures in the last column illustrate the effect of scaling up sample size (log scale) on the performance.

**Curse of big data phenomenon in molecular pre-training.**   We also conduct observations of scaling up the number of graphs on the downstream performance in molecular pre-training. We take AttrMasking [13], ContextPred [13], and GraphCL [39] as backbone models. Following their settings, the pre-training data is sampled from the molecular dataset ZINC15 and we take the molecular dataset BBBP as downstream data. We directly adopt the default settings of these pre-train models except for the number of molecules for pre-training. The number of molecular graphs are selected as {1,000,

5,000, 10,000, 50,000, 100,000}. The results are shown in Figure 5, which suggest that "curse of big data" phenomenon is still found even when pre-training and downstream data are all molecules.

Table 6: The value of parameters for fitting the curve according to the function $f(x) = a_1 \ln x / x^{a_2} + a_3$ $(a_1, a_2, a_3 > 0)$, based on the points in the last column in Figure 6 and Figure 7.

| Parameter / Dataset | GCC | | | GraphCL | | |
|---|---|---|---|---|---|---|
| | $a_1$ | $a_2$ | $a_3$ | $a_1$ | $a_2$ | $a_3$ |
| cora | 0.45 | 0 | 33.43 | 1038.19 | 1.24 | 17.15 |
| pubmed | 4.74 | 0.11 | 39.63 | 3.50 | 0.12 | 36.83 |
| brazil | 39.10 | 0.09 | 0 | 29.59 | 0.19 | 41.83 |
| dd242 | 6.60 | 0.08 | 3.82 | 7.84 | 0.12 | 0 |
| dd68 | 6.44 | 0.11 | 3.83 | 2.08e+17 | 6.55 | 10.37 |
| dd687 | 968.01 | 1.37 | 10.31 | 5.21 | 0.12 | 1.35 |
| wisconsin | 26.78 | 0.11 | 16.59 | 11.42 | 0.21 | 38.24 |
| cornell | 15.46 | 0.41 | 45.45 | 7.84 | 4.83 | 51.79 |
| imdb-binary | 7.43 | 0.13 | 64.97 | 1.06 | 0 | 51.69 |
| dd | 0.81 | 0.31 | 75.53 | 13.76 | 0.10 | 36.33 |
| msrc | 5.56 | 0.11 | 4.85 | 4.71 | 0.13 | 0 |

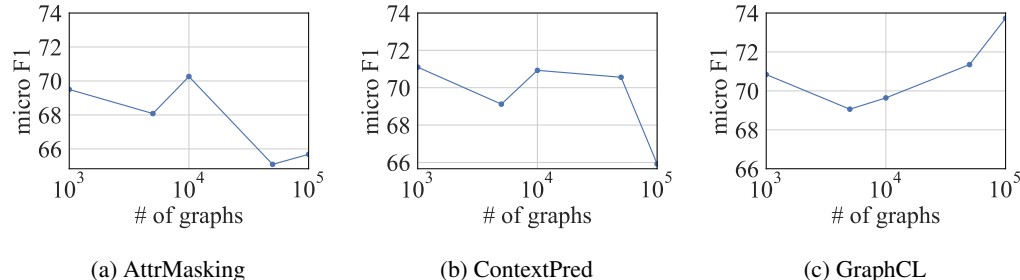

(a) AttrMasking     (b) ContextPred     (c) GraphCL

Figure 5: The effect of scaling up the number of graph datasets on the downstream performance under molecular pre-training on different pre-training models.

**The explanation of convex hull fit.** In order to better show the changing trend, the blue curve in the last column in Figure 6 and Figure 7 is fitted to the convex hull of the points. The convex hull is proposed to capture the performance of a randomized classifier made by choosing pre-training models with different probabilities [1].

We first introduce the concept of *randomized classifier*. Given two classifiers with training sample size and downstream performance $c_1 = (c_1^{sz}, c_1^{ds})$ and $c_2 = (c_2^{sz}, c_2^{ds})$, a *randomized classifier* can be made to choose the first classifier with probability $p$ and the second classifier with probability $1 - p$. Then the output of the randomized classifier is $pc_1 + (1 - p)c_2$, which is the convex combination of $c_1$ and $c_2$. All the points on this convex combination can be obtained by choosing different $p$. Extend the notion to the case of multiple classifiers, we can consider the output of such a randomized classifier to be a convex combination of the outputs of its endpoints [1]. All the points on the convex hull are achievable. Therefore, the output of the randomized classifier is equivalent to the convex hull of our trained classifiers' performance.

In our experiments, we include the upper hull of the convex hull of the model performances, *i.e.*, the highest downstream performance for every given sample size. Such convex hull fit is proved to be robust to the density of the points in each figure [1].

A final remark is that our observations on different downstream datasets do not result in a one-model-fits-all trend. So we propose to fit a complicated curve whose function has form $f(x) = a_1 \ln x / x^{a_2} + a_3$ $(a_1, a_2, a_3 > 0)$ to the best performing models (*i.e.*, the convex hull fit as discussed above). The fitted parameters $a_1$, $a_2$ and $a_3$ in this function of each curve are given in Table 6.

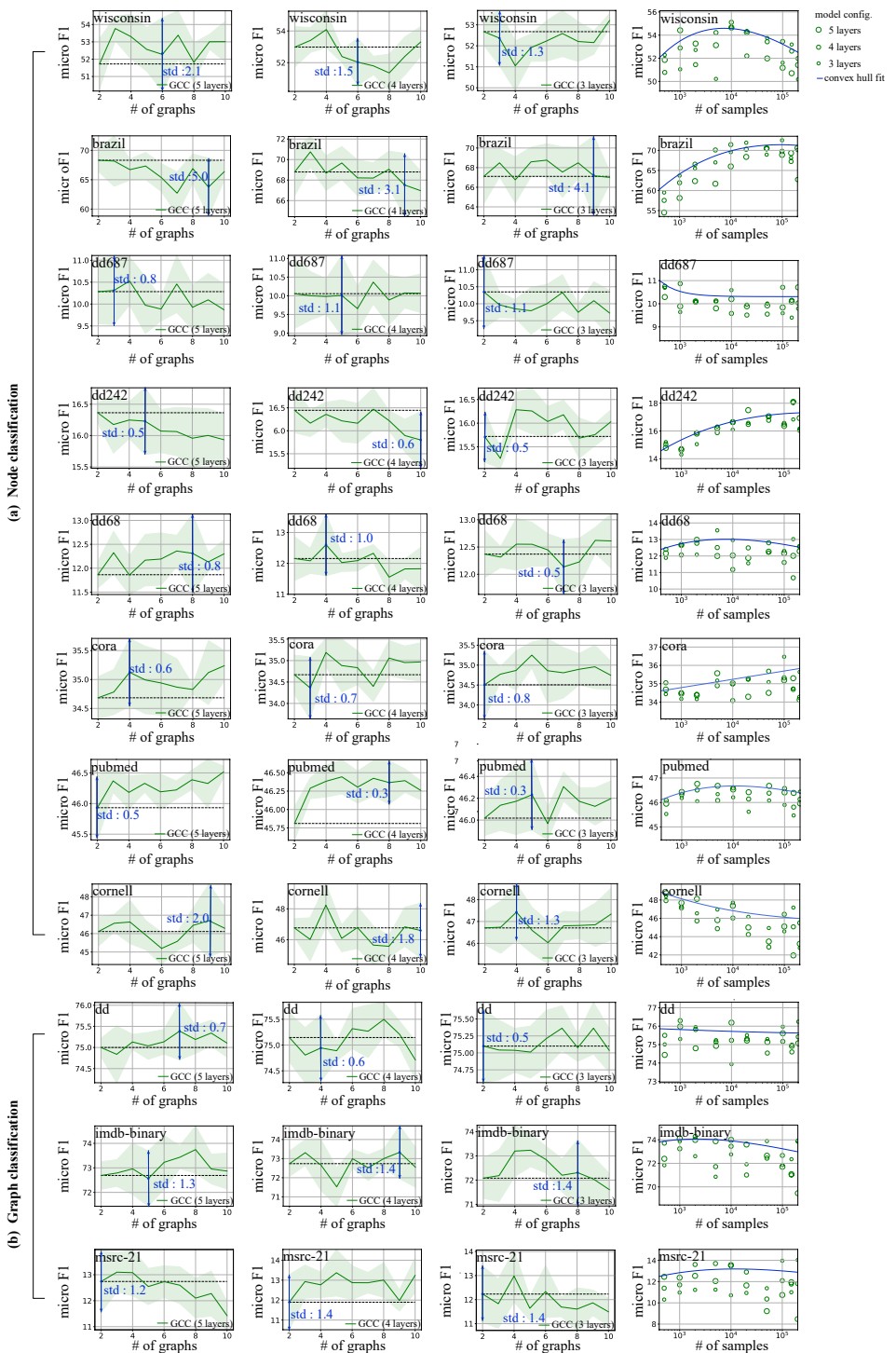

Figure 6: The additional observations of *curse of big data* phenomenon, performed on different GCC pre-training models. *Left three columns* present the effect of scaling up the number of graphs on the downstream performance under different model configurations (i.e., the number of GNN layers) respectively. *Last column* illustrates the effect of scaling up sample size (log scale) on the performance.

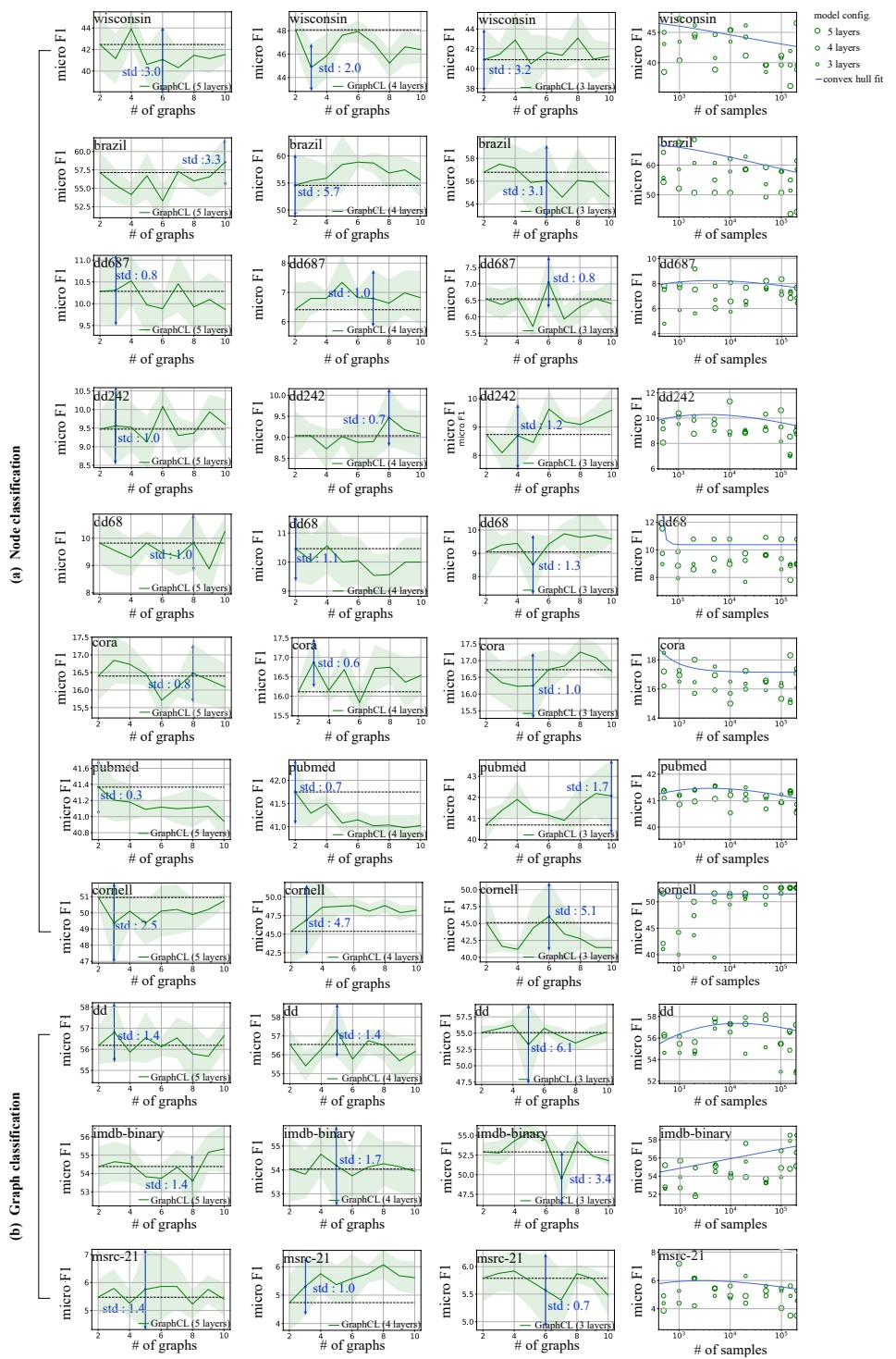

Figure 7: The additional observations of *curse of big data* phenomenon, performed on different GraphCL pre-training models.

# F  Empirical Study of Graph Properties

**Additional properties for part (b) in Figure 2.** In Figure 8, we plot the Pearson correlation between the graph properties of the graph used in pre-training (shown in the $y$-axis) and the performance of the pre-trained model using this graph on different unseen test datasets (shown in the $x$-axis). Note that the pre-training is performed on each of the input training graphs (in Table 3) via GCC. The results indicate that network entropy, density, average degree and degree variance exhibit a clear positive correlation with the performance, while the scale-free exponent presents an obviously negative relation with the performance. On the contrary, some other properties of graphs, including clique number, transitivity, degree assortativity and average clustering coefficient, do not seem to have connections with downstream performance, and also exhibit little or no correlation with the performance. Therefore, the favorable properties of network entropy, density, average degree, degree variance and the scale-free exponent of a real graph are able to characterize the contribution of a graph to pre-training.

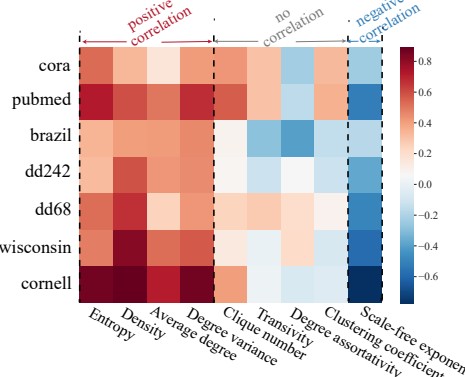

Figure 8: Pearson correlation between the structural features of the graph used in pre-training and the performance of the pre-trained model (using this graph) on different unseen test datasets.

**Detailed illustrations of Figure 3.**   In Figure 3, the illustrative graphs are generated by the configuration model with 15-18 nodes. The shaded area groups the illustrative graphs whose network entropy and graph properties are similar. Each four points on the same horizontal coordinate represent four graph properties of an illustrating graph. Each curve is fitted by least squares and represents the relation between entropy and other graph properties.

**Additional real-world example for Figure 3.**   In Figure 9, we provide a real-world example of how network entropy correlates with four typical structural properties (in red), as well as the performance of the pre-trained model on test graphs (in blue). Numerical experiments again support our explanation (or intuition) of their strong correlation.

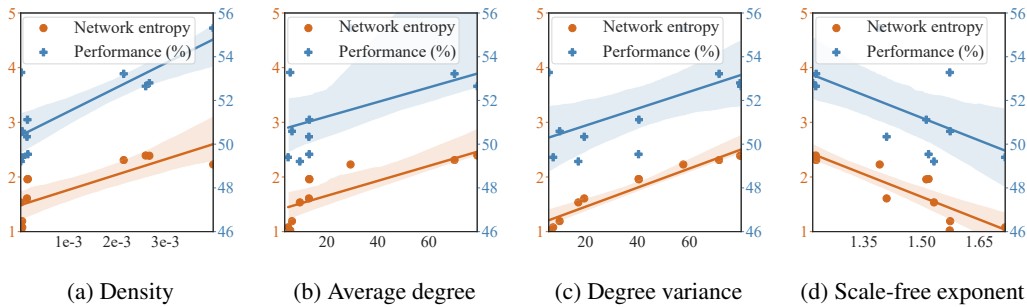

|  (a) Density  |  (b) Average degree  |  (c) Degree variance  |  (d) Scale-free exponent  |

Figure 9: The red plot shows the network entropy (left $y$-axis) versus typical structural properties in a graph (*i.e.*, density, average degree, degree variance, and the parameter $\alpha$ in a scale-free network), and the blue one shows the pre-training performances on wisconsin dataset (right $y$-axis) versus structural features.

# G   Implementation Details

The number reported in all the experiments are the mean and standard deviation over 10 evaluation results of the downstream task with random training/testing splits. When conducting the downstream task, For each dataset, we consistently use 90% of the data as the training set, and 10% as the testing set. We conduct all experiments on a single machine of Linux system with an Intel Xeon Gold 5118 (128G memory) and a GeForce GTX Tesla P4 (8GB memory).

**Implementations of our model.** The regularization for weights of the model in Eq. (5) is applied to first 2 layers of GIN. The maximal period of training one graph $F$ is 6, the maximum iteration number $T$ is 100, and the predictive uncertainty thresholds $T_s$ and $T_g$ are set to be 3 and 2 respectively. The selected instances are sampled from 20,000 instances each epoch. Since the pre-training model is unable to provide precise predictive uncertainty in the initial training stage, the model is warmed up over the first 20 iterations. Since we adopt GCC as the backbone pre-training model, the other settings are the same as GCC.

Our model is implemented under the following software settings: Pytorch version 1.4.0+cu100, CUDA version 10.0, networkx version 2.3, DGL version 0.4.3post2, sklearn version 0.20.3, numpy version 1.19.4, Python version 3.7.1.

**Implementations of baselines.** We compare against several graph representation learning methods. For implementation, we directly adopt their public source codes and most of their default hyperparameters. The key parameter settings and code links can be found in Table 7.

Table 7: The source code and major hyper-parameters used in the baselines.

| Method | Hyper-parameter | Code |
|---|---|---|
| DeepWalk | The dimension of output representations is 64, walk length = 10, number of walks = 80 | https://github.com/shenweichen/GraphEmbedding |
| struc2vec | The dimension of output representations is 32, walk length = 80, number of walks = 10, window size = 5 | https://github.com/leoribeiro/struc2vec |
| DGI | 512 hidden units per GNN layer, learning rate = 0.001 | https://github.com/PetarV-/DGI |
| GAE | 32 hidden units per GNN layer, learning rate = 0.01 | https://github.com/zfjsail/gae-pytorch |
| graph2vec | The dimension of output representations is 128 | https://github.com/benedekrozemberczki/graph2vec |
| InfoGraph | 32 hidden units per GNN layer, 5 layers | https://github.com/fanyun-sun/InfoGraph |
| DGCNN | 32 hidden units per GNN layer, learning rate = 0.001, batch size = 50 | https://github.com/leftthomas/DGCNN |
| GIN | 64 hidden units per GNN layer, 5 layers, learning rate = 0.01, sum pooling | https://github.com/weihua916/powerful-gnns |
| GraphCL | 300 hidden units per GNN layer, 5 layers, learning rate = 0.001 | https://github.com/Shen-Lab/GraphCL |
| GCC | 64 hidden units per GNN layer, 5 layers, learning rate = 0.005, number of samples per epoch = 20000 | https://github.com/THUDM/GCC |

# H   Additional Experimental Results and Analysis

**Analysis of ablation studies of APT.**   Table 1 presents the node classification results of APT and its 4 variants, *i.e.*, APT-G, APT-P, APT-R, and APT-L2. We here analyze the performance of these ablation studies from the following three points of view: (1) APT-L2 and APT, the variants with the proximity regularization w.r.t old knowledge, perform the best in most cases. This suggests that catastrophic forgetting of previous trained graphs could occur during pre-training, and it is necessary to add the proximity regularization to prevent this; (2) Sometimes APT-R (the variant without proximity term) achieves the best performance on the test data dd687 and cornell. One potential reason is that the last data to be selected for pre-training is dblp, while the test data dd687 and cornell exhibit some properties similar to dblp. Therefore, even excluding the proximity term, the pre-training model can still remember knowledge from the last pre-trained data, and achieve good performance on the test data that is similar to the last pre-trained data; (3) APT-G and APT-P present suboptimal performance among all variants, which suggests the utility of graph properties and predictive uncertainty.

**Impact of five graph properties combination.**   We study the effect of the strategy of utilizing only one graph property in Table 8. We find that the five properties used in our model are all indispensable, and the most important one probably varies for different datasets. That's why we choose to combine all graph properties.

Moreover, these case studies may provide some clues of how to select pre-training graphs when some knowledge of the downstream tasks is known. For example, if the downstream dataset is extremely dense (like imdb-binary), the density property dominates among the selection criteria (such that the probability of encountering very dense out-of-distribution samples during testing can be reduced). If the entropy of downstream dataset is very high (like brazil), it is perhaps better to choose graphs with high entropy for pre-training. But still, when the downstream task is unknown, using the combination of five metrics often leads to the most satisfactory and robust results.

Table 8: The effect of different graph properties on downstream performance (micro F1 is reported) under APT-L2 (fine-tune). The last row is our strategy of combining all the graph properties, and each of the first five rows is the strategy of only utilizing one graph property.

| Dataset / Method | node classification | | | | | | | | graph classification | | |
|---|---|---|---|---|---|---|---|---|---|---|---|
| | brazil | dd242 | dd68 | dd687 | wisconsin | cornell | cora | pubmed | imdb-binary | dd | msrc-21 |
| Entropy | **80.04(2.15)** | 25.79(0.94) | 16.31(0.81) | 11.08(0.82) | 67.01(2.00) | 52.80(2.46) | 45.41(0.85) | 50.85(0.19) | 76.78(0.84) | 75.56(0.84) | 24.34(1.50) |
| Density | 79.23(1.92) | **27.29(0.62)** | **19.89(0.95)** | 12.22(1.06) | 65.58(1.87) | 51.15(1.59) | 46.18(0.71) | 50.74(0.15) | **77.20(0.66)** | 75.29(0.54) | 24.20(1.31) |
| Average degree | 79.22(1.65) | 24.99(0.67) | 16.56(1.01) | 11.67(1.18) | 67.02(1.86) | 51.43(4.16) | 46.38(0.48) | 50.99(0.31) | 76.87(0.93) | 75.46(0.53) | 25.14(1.54) |
| Degree variance | 78.44(2.24) | 24.94(0.61) | 16.62(1.04) | 11.51(1.17) | 65.65(1.28) | 50.45(2.14) | 45.76(0.65) | 50.70(0.21) | 76.39(1.04) | 75.47(0.67) | 25.22(1.51) |
| Scale-free exponent | 79.70(2.71) | 24.94(0.68) | 17.26(0.63) | 12.03(1.41) | 64.77(2.31) | 51.37(2.70) | 45.18(0.52) | 50.84(0.26) | 75.24(0.62) | 75.52(1.24) | 23.19(1.39) |
| Our combination | 78.75(1.63) | 24.62(0.90) | 17.83(1.35) | **12.26(0.78)** | **67.04(1.50)** | **52.94(1.95)** | **47.48(0.46)** | **51.25(0.21)** | 75.93(0.84) | **75.58(1.06)** | **25.58(1.57)** |

**Analysis of the selected graphs.**   The data sequentially selected via our graph selector are uillinois, soc-sign0811, msu, michigan, wiki-vote, soc-sign0902 and dblp. To further analyze why these graphs are chosen, we present their detailed structural properties in Table 4 in the Appendix D. We first observe that uillinois, michigan and msu have the largest value of the term related to graph properties (*i.e.*, $\mathrm{MEAN}(\hat{\phi}_{\mathrm{entropy}}, \hat{\phi}_{\mathrm{density}}, \hat{\phi}_{\mathrm{avg\_deg}}, \hat{\phi}_{\mathrm{deg\_var}}, -\hat{\phi}_{\alpha})$), while dblp has the smallest. This indicates that the selection of data is influenced by a combination of factors, including graph properties and potentially other factors such as predictive uncertainty. Moreover, it is also interesting to see that the selected graph wiki-vote is the smallest graph among all the pre-training graphs, but it still contributes to the performance. This observation again verifies the curse of big data phenomenon in graph pre-training, which suggests that having more training samples does not necessarily result in improved downstream performance.

**Results of APT under different backbone models.**   We here include GraphCL [39] and JOAO [38] as our backbone models. Table 9 shows our superiority under most cases. The datasets used for pre-training and testing are the same as those employed in the experiments where GCC was the backbone model (refer to Table 7). Since the downstream tasks of GraphCL and JOAO are limited to graph classification, we can directly adapt them to our setting with graph classification as downstream task. For the node classification task, we simply take the RWR subgraphs as the input of GraphCL and JOAO, and treat the learned subgraph representation as node representation. We directly adopt GraphCL and JOAO with their default hyper-parameters. The backbone GNN model is GIN with 5 layers.

Table 9: Micro F1 scores of different models in the node classification task and graph classification task, with GraphCL and JOAO as our backbone model, respectively.

| Method \ Dataset | node classification | | | | | | | | graph classification | | |
|---|---|---|---|---|---|---|---|---|---|---|---|
| | brazil | dd242 | dd68 | dd687 | wisconsin | cornell | cora | pubmed | imdb-binary | dd | msrc-21 |
| GraphCL (rand, finetune) | 30.45(0.37) | 40.73(0.66) | 64.43(14.95) | 15.04(0.85) | **14.69(2.48)** | **10.99(0.58)** | 63.85(2.18) | 44.21(10.58) | 63.60(3.61) | 58.15(4.60) | 8.25(2.94) |
| GraphCL (fine-tune) | **30.81(0.36)** | 42.91(0.91) | 73.57(10.33) | 15.35(0.99) | 13.51(2.57) | 10.66(1.04) | 63.85(4.42) | 51.05(2.41) | 66.90(4.39) | 65.55(5.14) | 8.77(2.60) |
| GraphCL-APT (fine-tune) | 30.63(0.49) | **42.93(1.01)** | **75.14(10.02)** | **15.42(1.49)** | 14.51(2.72) | 10.88(0.84) | **64.46(3.22)** | **52.63(2.63)** | **67.55(3.37)** | **67.23(4.07)** | **10.02(2.78)** |
| JOAO (rand, finetune) | 29.93(2.84) | 42.01(0.68) | 72.14(6.74) | **10.93(2.85)** | 8.08(2.15) | 7.40(3.48) | 45.38(13.30) | 45.26(10.31) | 67.70(3.35) | 62.10(4.31) | 11.40(3.06) |
| JOAO (fine-tune) | 29.34(3.04) | 42.21(0.88) | **75.00(5.76)** | 10.54(3.07) | 7.56(1.94) | 8.77(2.39) | 50.0(12.28) | 42.11(10.26) | **68.50(3.61)** | 62.61(4.99) | 10.18(1.72) |
| JOAO-APT (fine-tune) | **30.45(0.56)** | **42.80(1.05)** | 72.29(11.03) | 10.71(2.85) | **9.32(4.32)** | **10.90(0.67)** | **51.92(1.98)** | **46.51(2.69)** | 63.9(3.48) | **63.42(3.61)** | **12.82(0.70)** |

**Results of APT under molecular pre-training.** We also evaluate APT under molecular pre-training setting. We adopt the state-of-art model, Mole-BERT [35] and a commonly used strategy, EdgePred [12] as backbone molecular pre-training models.

The predictive uncertainty mentioned before is based on contrastive loss, however, the loss function of Mole-BERT and EdgePred are different from that used in GCC. To adapt these two models, we need to make corresponding changes to our predictive uncertainty. More specifically, for Mole-BERT, we take triplet masked contrastive learning $\mathcal{L}_{\text{TMCL}}$, which is the combination of triplet loss $\mathcal{L}_{\text{tri}}$ with commonly-used contrastive loss $\mathcal{L}_{\text{con}}$ in the original paper, as the predictive uncertainty. For EdgePred, we take the loss function, *i.e.*, negative log likelihood in the original paper as the predictive uncertainty

Besides, we adjust the data selection process in molecular pre-training. This is because each molecular graph is taken as a sample in molecular pre-training, so we can directly select graphs. The overall algorithm for APT under molecular setting is given in Algorithm 2.

---

**Algorithm 2** Overall algorithm for APT (Mole).

---

**Input:** A collection of graphs $\mathcal{G} = \{G_1, \ldots, G_N\}$, the number of graphs in the pre-training dataset $|\mathcal{G}^0|$, the maximal period $F$ of training a set of graphs, trade-off parameter $\gamma_t = 0$, ratio of the number of selected graphs each time to the number of graphs in the pre-training dataset $\eta$, hyperparameter $\{\beta_t\}$, the learning rate $\mu$, the predictive uncertainty threshold of moving to a new set of graphs $T_g$, and the maximum iteration number $T$.
**Output:** Model parameter $\theta$ of the pre-trained graph model.

1: Choose a set of graphs $\mathcal{G}^*$ ($|\mathcal{G}^*| = \eta * |\mathcal{G}^0|$) from $\mathcal{G}$ via the graph selector, and $\mathcal{G} \leftarrow \mathcal{G}\backslash\{\mathcal{G}^*\}$.
2: **while** The iteration number reaches $T$ **do**
3:    Update model parameters $\theta \leftarrow \theta - \mu\nabla_\theta\mathcal{L}(\theta)$.
4:    **if** $\phi_{\text{uncertain}}(\mathcal{G}^*) < T_g$ *or* the model has been trained on $\mathcal{G}^*$ by $F$ iterations **then**
5:       Update the trade-off parameter $\gamma_t \sim \text{Beta}(1, \beta_t)$.
6:       Choose a set of graphs $\mathcal{G}^*$ ($|\mathcal{G}^*| = \eta * |\mathcal{G}^0|$) from $\mathcal{G}$, and $\mathcal{G} \leftarrow \mathcal{G}\backslash\{\mathcal{G}^*\}$.
7:    **end if**
8: **end while**

---

For pre-training datsets, we use 2 million molecules sampled from the ZINC15 database following [12, 35]. For downstream datasets, we adopt the widely-used 8 binary classification datasets contained in MoleculeNet [33] following [12, 35]. For implementation, $T_g$ is set as -2.5 for Mole-BERT and 0.5 for EdgePred. For both Mole-BERT and EdgePred, the ratio of the number of selected graphs each time to the number of the graphs in original pre-training dataset $\eta$ is 0.2. Other parameters related to APT are set the same as those in APT with GCC as pre-training backbone model. The other parameters related to molecular pre-training are the same as those employed in Mole-BERT and EdgePred, respectively.

The results of APT under molecular pre-training with Mole-BERT and EdgePred as backbone models are reported in Table 10. We can see that our APT adaptions show significant superiority compared to the backbones. Moreover, APT (Mole-BERT) achieves the best performance on 6 out of 8 downstream datasets among all the baseline models, which shows the effectiveness of our pipeline.

**Training time.** As empirically noted in Table 11, the total training time of APT-L2 and APT is 18321.39 seconds and 18592.01 seconds respectively (including the time consumed in graph selection

Table 10: Results for molecular property classification. We report the mean (standard deviation) ROC-AUC of 10 random seeds with scaffold splitting. 'No pre-train' means training from scratch. The baseline results are directly obtained from the reported results in Mole-BERT [35]..

| Dataset / Method | Tox21 | ToxCast | Sider | ClinTox | MUV | HIV | BBBP | Bace | Average |
|---|---|---|---|---|---|---|---|---|---|
| # Molecules | 7,831 | 8,575 | 1,427 | 1,478 | 93,087 | 41,127 | 2,039 | 1,513 | - |
| No pretrain | 74.6(0.4) | 61.7(0.5) | 58.2(1.7) | 58.4(6.4) | 70.7(1.8) | 75.5(0.8) | 65.7(3.3) | 72.4(3.8) | 67.15 |
| InfoGraph | 73.3(0.6) | 61.8(0.4) | 58.7(0.6) | 75.4(4.3) | 74.4(1.8) | 74.2(0.9) | 68.7(0.6) | 74.3(2.6) | 70.10 |
| GPT-GNN | 74.9(0.3) | 62.5(0.4) | 58.1(0.3) | 58.3(5.2) | 75.9(2.3) | 65.2(2.1) | 64.5(1.4) | 77.9(3.2) | 68.45 |
| ContextPred | 73.6(0.3) | 62.6(0.6) | 59.7(1.8) | 74.0(3.4) | 72.5(1.5) | 75.6(1.0) | 70.6(1.5) | 78.8(1.2) | 70.93 |
| GraphLoG | 75.0(0.6) | 63.4(0.6) | 59.6(1.9) | 75.7(2.4) | 75.5(1.6) | 76.1(0.8) | 68.7(1.6) | 78.6(1.0) | 71.56 |
| G-Contextual | 75.0(0.6) | 62.8(0.7) | 58.7(1.0) | 60.6(5.2) | 72.1(0.7) | 76.3(1.5) | 69.9(2.1) | 79.3(1.1) | 69.34 |
| G-Motif | 73.6(0.7) | 62.3(0.6) | 61.0(1.5) | 77.7(2.7) | 73.0(1.8) | 73.8(1.2) | 66.9(3.1) | 73.0(3.3) | 70.16 |
| AD-GCL | 74.9(0.4) | 63.4(0.7) | 61.5(0.9) | 77.2(2.7) | 76.3(1.4) | 76.7(1.2) | 70.7(0.3) | 76.6(1.5) | 72.16 |
| JOAO | 74.8(0.6) | 62.8(0.7) | 60.4(1.5) | 66.6(3.1) | 76.6(1.7) | 76.9(0.7) | 66.4(1.0) | 73.2(1.6) | 69.71 |
| SimGRACE | 74.4(0.3) | 62.6(0.7) | 60.2(0.9) | 75.5(2.0) | 75.4(1.3) | 75.0(0.6) | 71.2(1.1) | 74.9(2.0) | 71.15 |
| GraphCL | 75.1(0.7) | 63.0(0.4) | 59.8(1.3) | 77.5(3.8) | 76.4(0.4) | 75.1(0.7) | 67.8(2.4) | 74.6(2.1) | 71.16 |
| GraphMAE | 75.2(0.9) | 63.6(0.3) | 60.5(1.2) | 76.5(3.0) | 76.4(2.0) | 76.8(0.6) | 71.2(1.0) | 78.2(1.5) | 72.30 |
| 3D InfoMax | 74.5(0.7) | 63.5(0.8) | 56.8(2.1) | 62.7(3.3) | 76.2(1.4) | 76.1(1.3) | 69.1(1.2) | 78.6(1.9) | 69.69 |
| GraphMVP | 74.9(0.8) | 63.1(0.2) | 60.2(1.1) | 79.1(2.8) | 77.7(0.6) | 76.0(0.1) | 70.8(0.5) | 79.3(1.5) | 72.64 |
| MGSSL | 75.2(0.6) | 63.3(0.5) | 61.6(1.0) | 77.1(4.5) | 77.6(0.4) | 75.8(0.4) | 68.8(0.6) | 78.8(0.9) | 72.28 |
| AttrMask | 75.1(0.9) | 63.3(0.6) | 60.5(0.9) | 73.5(4.3) | 75.8(1.0) | 75.3(1.5) | 65.2(1.4) | 77.8(1.8) | 70.81 |
| MAM (with vanilla VQ-VAE) | 75.8(0.6) | 63.1(0.5) | 60.7(1.5) | 75.4(2.7) | 76.5(1.6) | 76.2(0.9) | 66.4(0.7) | 78.2(0.8) | 71.39 |
| TMCL( w/o $\mathcal{L}_{con}$ ) | 73.5(1.0) | 61.8(0.3) | 58.7(1.6) | 61.1(4.1) | 71.6(1.3) | 73.5(1.3) | 65.4(2.6) | 73.7(2.4) | 67.41 |
| TMCL ( w/o $\mathcal{L}_{tri}$ ) | 74.1(0.4) | 62.4(0.8) | 58.7(3.0) | 75.6(2.2) | 75.7(1.1) | 74.6(1.1) | 66.8(1.4) | 74.2(1.3) | 70.26 |
| MAM | 76.2(0.5) | 63.9(0.3) | 61.4(1.9) | 75.1(3.0) | 77.4(2.1) | 77.5(1.0) | 66.8(1.5) | 78.9(1.1) | 72.16 |
| TMCL | 74.9(0.7) | 63.2(0.7) | 59.6(1.4) | 77.0(4.2) | 77.2(0.3) | 75.3(1.1) | 67.6(1.3) | 75.1(1.2) | 71.24 |
| EdgePred | 76.0(0.6) | 64.1(0.6) | 60.4(0.7) | 64.1(3.7) | 75.1(1.2) | 76.3(1.0) | 67.3(2.4) | 77.3(3.5) | 70.08 |
| APT (EdgePred) | 76.5 (0.4) | 64.3 (0.3) | 61.4 (0.8) | 77.9 (3.0) | 75.2 (1.7) | 76.6 (0.9) | 70.7 (2.0) | 81.8 (1.4) | 73.05 |
| Mole-BERT | 76.8(0.5) | 64.3(0.2) | **62.8(1.1)** | 78.9(3.0) | **78.6(1.8)** | 78.2(0.8) | 71.9(1.6) | 80.8(1.4) | 74.04 |
| APT (Mole-BERT) | **77.0(0.4)** | **65.0(0.3)** | 60.8(0.4) | **79.9(1.0)** | 75.9(1.0) | **78.9(0.6)** | **73.1(0.4)** | **82.3(1.4)** | **74.11** |

and regularization term), while the competitive graph pre-training model GCC takes 40161.68 seconds for the same number of training epochs on the same datasets.

- The time spent on the inference on all graphs during graph selection (which is the main time spent for graph selection) only accounts for 3.95% and 3.87% of the total time under APT-L2 and APT respectively. Note that this step is executed only in a few epochs (around 6% in our current model).

- The time cost of the L2 regularization term only accounts for 0.08% of the total time and the EWC regularization term only accounts for 0.45% of the total time, which is calculated by the runtime gap between the models with and without the regularization term. Note that the regularization term is imposed on the first two layers of the GNN encoder, which only accounts for 12.4% of the total number of parameters.

The efficiency of our model is due to a much smaller number of carefully selected training graphs and samples at each epoch. In addition, the number of parameters in our model is 190,544, which is the same order of magnitude as classical GNNs like GraphSAGE, GraphSAINT, *etc.* and is relatively small among models in open graph benchmark [11].

Table 11: Training time (sec) comparison between our model and GCC. All the models are trained under the same number of epochs, which is set as 100 in practice. (The difference in time cost of inference on all graphs is due to different runs.)

| Time | GCC | APT-L2 | APT |
|---|---|---|---|
| inference on all graphs | - | 723.92 | 719.64 |
| proximity term | - | 15.98 | 83.58 |
| **total** | 40161.68 | 18321.39 | 18592.01 |

**Effects of hyper-parameters.** Here we show that the effect of the hyper-parameter $\{\lambda\}$ in the proximity term, the maximal number of training iterations on one graph $F$, the predictive uncertainty threshold of moving to a new graph $T_g$, and the predictive uncertainty threshold of choosing training samples $T_s$ on performance in Figure 10 and Figure 11.

The hyper-parameter $\lambda$ is the trade-off parameters between the knowledge learnt from new data and that from previous data in Eq. (5). We use the dataset dd242 as an example to find the suitable values of the hyper-parameter under the L2 and EWC regularization setting respectively, and present here for reference (see Figure 10). Clearly, a too small or too large $\lambda$ would deteriorate the performance. Thus, an appropriate value of $\lambda$ is preferred to ensure that the graph pre-training model can learn from new data as well as remember previous knowledge. We leave changing $\lambda$ as the future work.

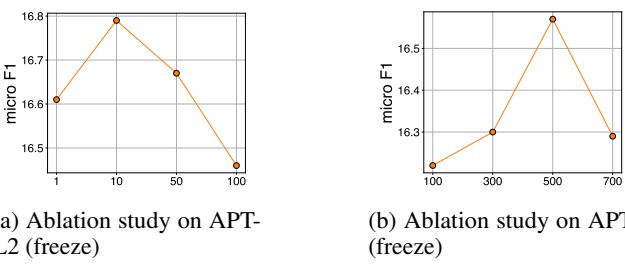

(a) Ablation study on APT-L2 (freeze)

(b) Ablation study on APT (freeze)

Figure 10: Performance of our model on dd242 w.r.t varying $\{\lambda\}$.

Our model training involves three hyper-parameter $F, T_g, T_s$, where $F$ controls the largest number of epochs training on each graph, $T_g$ is the predictive uncertainty threshold of moving to a new graph, $T_s$ is the predictive uncertainty threshold of choosing training samples. We use grid search to show $F \in \{4, 5, 6\}$'s, $T_g \in \{3, 3.5, 4\}$'s and $T_s \in \{1, 2, 3\}$'s role in the pre-training. $F$ remains at 5 while studying $T_g$ and $T_s$, $T_g$ remains at 3.5 while studying $F$ and $T_s$, and $T_s$ remains at 2 while studying $F$ and $T_g$. Figure 11 presents the effect of these parameters, We find that if the value of $F$ is set too small or that of $T_g$ is too large, the model cannot learn sufficient knowledge from each graph, leading to suboptimal results. Too large $F$ or small $T_g$ also lead to poor performance. This indicates that instead of training on a graph for a large period, it would be better to switch to training on various graphs in different domains to gain diverse and comprehensive knowledge. Regarding the hyper-parameter $T_s$, we observe that large $T_s$ would make the model having too few training samples to learn knowledge, and small $T_s$ could not select the most uncertain and representative samples, thus both cases achieve suboptimal performance.

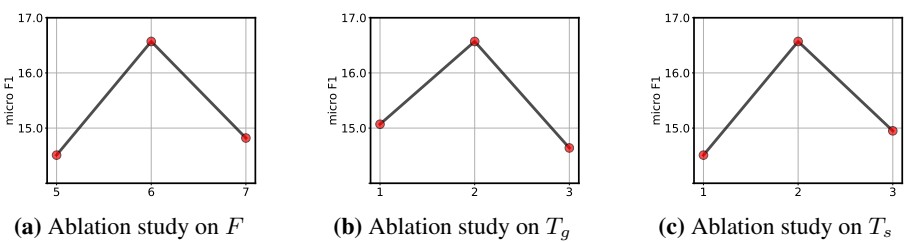

**(a)** Ablation study on $F$      **(b)** Ablation study on $T_g$      **(c)** Ablation study on $T_s$

**Figure 11: Performance of our model on dd242 w.r.t varying $F, T_g, T_s$.**

**The choice of $\beta_t$, its alternatives, and ablation study.** At the beginning of the pre-training, the model is less accurate and needs more guidance from graph properties. We therefore set $\gamma_t$ as larger at the beginning and gradually decrease it. To simplify this process, we follow [3] to use the exponential formula of $\beta_t = c_1 - c_2^t$ to set the expectation of $\gamma_t$ to be strictly decreasing (where $\gamma_t \sim \text{Beta}(1, \beta_t)$).

The parameters $c_1$ and $c_2$ in the exponential formula of $\beta_t = c_1 - c_2^t$ are suggested as 1.005 and 0.995 in [3]. We simply perform grid search on $c_1$ in $\{1.005, 3, 5\}$; see the effect of $c_1$ in the Figure 12.

We then illustrate that the choice of the decay function of $\beta_t$ is robust. Table 12 and Table 13 below show the effect of linear decay, step decay and exponential decay on $\beta_t$. (The function for linear decay and step decay are designed as $\beta_t = 2.001 + 0.004t$, $\beta_t = 2.005 + \text{floor}(t/20)$, respectively. The initial value $\beta_1$ is set the same as ours.) While there is no universally better decay function, the performance of our method is not significantly impacted by the choice of different decay functions, and our performance is better than the baselines in most cases regardless of the choices of specific decay functions.

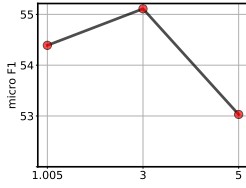
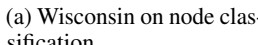
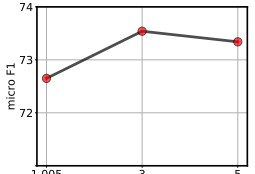

(a) Wisconsin on node classification

(b) Imdb-binary on graph classification

Figure 12: Performance of APT-L2 (freeze) w.r.t varying $c_1$.

Table 12: Micro F1 of APT-L2 (freeze) with the different decay functions in the node classification task.

| Method \ Dataset | brazil | dd242 | dd68 | dd687 | wisconsin | cornell | cora | pubmed |
|---|---|---|---|---|---|---|---|---|
| linear | **72.30(1.37)** | 16.28(0.57) | 12.44(0.72) | 10.29(0.87) | 54.20(1.50) | 47.66(1.53) | **35.74(0.52)** | 46.49(0.19) |
| step | 68.70(3.95) | 16.74(0.45) | **12.86(1.07)** | 10.09(0.76) | 52.55(2.39) | 48.08(1.28) | 35.50(0.46) | **46.58(0.21)** |
| exponential | 69.82(2.32) | **16.79(0.88)** | 12.68(0.81) | **10.34(1.12)** | **55.11(1.74)** | **48.76(2.20)** | 34.27(0.43) | 46.21(0.15) |

Table 13: Micro F1 of APT-L2 (freeze) with the different decay functions in the graph classification task.

| Method \ Dataset | imdb-binary | dd | msrc-21 |
|---|---|---|---|
| linear | **73.66(0.34)** | 75.47(0.26) | 13.01(0.78) |
| step | 72.99(0.40) | 75.41(0.41) | **14.13(0.56)** |
| exponential | 73.54(0.40) | **75.81(0.30)** | 13.16(0.77) |

**Results under GCC's original experimental setting.** For a more comprehensive comparison, we also adopt the same pretraining datasets and downstream datasets as GCC. Table 14 and table 15 shows the performance of our model and the strongest competitor GCC, under the experiment setting of GCC in node classification and graph classification task, repectively. (The performance of GCC is directly taken from its paper.) The results indicate that our model still outperforms GCC under the experimental setting of GCC. The results still suggest ours superiority in most cases.

Table 14: Micro F1 scores of GCC and our model in the node classification task, under the experimental setting of GCC.

| Method \ Dataset | US-Airport | H-index |
|---|---|---|
| GCC (freeze) | 65.6 | 75.2 |
| Ours (freeze) | **66.12(5.31)** | **75.9(3.01)** |
| GCC (fine-tune) | 67.2 | 80.6 |
| Ours (fine-tune) | **70.50(6.08)** | **82.28(1.48)** |

Table 15: Micro F1 scores of GCC and our model in the graph classification task, under the experimental setting of GCC.

| Method \ Dataset | IMDB-B | IMDB-M | COLLAB | RDT-B | RDT-M |
|---|---|---|---|---|---|
| GCC (freeze) | 72.0 | 49.4 | 78.9 | 89.8 | 53.7 |
| Ours (freeze) | **73.2(0.27)** | **49.6(1.17)** | **79.12(0.77)** | 89.6(2.51) | **54.1(2.45)** |
| GCC (fine-tune) | 73.8 | 50.3 | 81.1 | 87.6 | 53.0 |
| Ours (fine-tune) | **76.27(1.20)** | **50.50(1.08)** | **81.23(0.86)** | **92.20(2.43)** | **53.28(2.33)** |

**The justification of input graphs' learning order.**  Table 16 reveals the downstream performance can be affected by the learning order of input training graphs. With the guidance of graph selector, the pre-training model is encouraged to first learn the graphs and samples with higher predictive uncertainty and graph properties. Such learning order accomplishes better downstream performance compared to the reverse or random one.

Table 16: The effect of input graphs' learning order on downstream performance (micro F1 is reported) under freezing mode in the node classification task. The first row is the order learnt from APT-L2, and the second and third rows are the reverse and random order of the first row, respectively.

| Method \ Dataset | brazil | dd242 | dd68 | dd687 | wisconsin | cornell | cora | pubmed |
|---|---|---|---|---|---|---|---|---|
| Our order | **69.82(2.32)** | **16.79(0.88)** | **12.68(0.81)** | 10.34(1.12) | **55.11(1.74)** | **48.76(2.20)** | 34.27(0.43) | 46.21(0.15) |
| Reverse order | 69.60(2.71) | 16.00(0.47) | 11.41(0.91) | 10.65(0.65) | 51.46(1.64) | 44.36(1.38) | 35.66(0.62) | 45.92(0.14) |
| Random order | 67.25(2.40) | 16.11(0.79) | 12.57(1.17) | **11.06(0.75)** | 53.06(2.41) | 46.76(1.95) | **35.90(0.72)** | **46.36(0.20)** |

**The choice of the "difficult" data.**  Among all the data, "difficult" samples contribute the most to the loss function, and thus they can yield gradients with large magnitude. Comparatively, training with easy samples may suffer from inefficiency and poor performance as these data points produce gradients with magnitudes close to zero [14, 30]. In addition, learning from difficult samples has proven to be able to accelerate convergence and enhance the expressive power of the learnt representations [28, 31]. For our model, the importance of learning from difficult samples is also justified empirically, as shown in Table 17.

Table 17: The comparison of learning from easy samples and learning from difficult sample in our pipeline (APT-L2 (freeze)) on node classification. Micro F1 is reported in the table. (Under the setting of learning from easy samples, we replace $\phi_{\text{uncertain}}$ with $-\phi_{\text{uncertain}}$ in Eq.(4), and only sample instances with predictive uncertainty lower than $T_s$.)

| Method \ Dataset | brazil | dd242 | dd68 | dd687 | wisconsin | cornell | cora | pubmed |
|---|---|---|---|---|---|---|---|---|
| Learning from easy samples | 56.34(3.45) | 14.38(0.53) | 11.76(1.04) | 9.90(0.64) | 50.65(1.84) | 48.09(1.72) | **35.74(0.42)** | 46.03(0.17) |
| Learning from difficult samples (ours) | **69.82(2.32)** | **16.79(0.88)** | **12.68(0.81)** | **10.34(1.12)** | **55.11(1.74)** | **48.76(2.20)** | 34.27(0.43) | **46.21(0.15)** |

**Time comparison: pre-training vs. training from scratch.**  Using a pre-trained model can significantly reduce the time required for training from scratch. The reason is that the weights of the pre-trained model have already been put close to appropriate and reasonable values; thus the model converges faster during fine-tuning on a test data. As shown in Figure 13, compared to regular GNN model (*e.g.* GIN), our model yields a speedup of 4.7× on average (which is measured by the ratio of the training time of GIN to the fine-tuning time of APT). Based on above analysis, we can draw a conclusion that pre-training is beneficial both in effectiveness and efficiency.

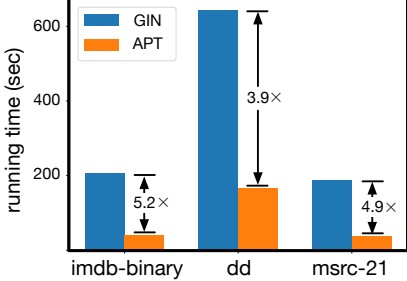

Figure 13: The running time of our model and the basic GNN model on graph classification task. Our model achieves a speedup of 4.7× on average compared with GIN.

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
