# OpenReview forum: "Better with Less: A Data-Active Perspective on Pre-Training Graph Neural Networks"
_NeurIPS.cc/2023/Conference — NeurIPS 2023 poster_

### Official Review · Reviewer_gzMz · 2023-06-14

**Soundness:** 2 fair
**Presentation:** 2 fair
**Contribution:** 3 good
**Rating:** 6
**Confidence:** 3

**Summary:**

This work found that more data does not necessarily improve the pre-training of GNNs and then propose a method to adaptively select data for pre-training GNNs. Experimental results show the proposed method is effective.

**Strengths:**

1. this work is well motivated by the observation that more data does not necessarily lead to better pretrained graph encoder

2. that the experiments seems to be convincing since APT generally outperforms other baselines

**Weaknesses:**

There are several weaknesses mainly about the method:

1. The optimization problem in equation 5 is confusing, what is the variable you are optimizing, a series of binary variable indicating whether to select a graph or not? Based on description in subsequent part, you are basically selecting graphs with the highest score, in that case it is unnecessary to formulate an optimization problem, just state your scoring function.

2. The description is confusing, eg, what is M in line 296? The number of subgraphs being selected?

3. Efficiency is important in data selection, but the authors do not discuss it in the main body of the paper. I assume the method is slower than baseline pretraining method but how slow is it?

4. The motivation of the other four property criteria is unclear, especially when they are highly correlated with the network entropy. Can we just drop the other four criteria?

5. The authors claim in the conclusion that APT can enhance the model with a fewer number of input data, but I can't find evidence for that. Is APT only trained on fewer data than baselines?

Overall, the description of the proposed method could be improved. In addition, the authors claimed they are not curriculum learning but I don't think so. Time-adaptive selection strategy is just curriculum learning in a broader sense, I think easy-first is just one type of curriculum.

**Questions:**

See weakness above

**Limitations:**

The authors do not state limitations in either the appendix or the main body of the paper, please add them.

---

> ### Author Rebuttal · Authors · 2023-08-06
>
> Dear Reviewer gzMz,
>
> We really appreciate your valuable comments. We have followed closely the suggestions, and made clarifications and revisions accordingly. We hope the revised content could help to further strengthen our work.
>
> >W1: The optimization problem in equation 5 is confusing, what is the variable you are optimizing, a series of binary variable indicating whether to select a graph or not? Based on description in subsequent part, you are basically selecting graphs with the highest score, in that case it is unnecessary to formulate an optimization problem, just state your scoring function.
>
> Sorry for the confusion. We here aim to select graphs with the highest score $\mathcal J(G)$. As suggested, we have removed the optimization problem and modified the description to: "Thus we can select the graph with the highest score, where the score is defined as $\mathcal{J}(G) = \gamma_t \phi_\mathrm{uncertain} +  (1 - \gamma_t)\mathrm{MEAN}(\hat{ \phi} _ \mathrm{entropy}, \hat{\phi}_\mathrm{density}, \hat{\phi} _\mathrm{avg\\_deg}, \hat{\phi} _\mathrm{deg\\_var},- \hat{\phi} _\mathrm{\alpha}) $.
> >W2: The description is confusing, eg, what is M in line 296? The number of subgraphs being selected?
>
> Thanks for pointing it out. $M$ is a hyper-parameter used to compute the graph-level predictive uncertainty $\phi_{\text{uncertain}}(G) = (1/M) \sum_{i=1}^M \mathcal L_i$, where $M$ denotes the number of subgraph instances queried in the graph $G$ for uncertainty estimation (as mentioned in line 126). We have provided clear descriptions of the meaning of $M$ in line 296, and other ambiguous terms or concepts in our revised version.
>
> >W3: Efficiency is important in data selection, but the authors do not discuss it in the main body of the paper. I assume the method is slower than baseline pre-training method but how slow is it?
>
> The training time comparison between our model and the most competitive pre-training baseline GCC have been provided in Table 10 in Appendix H. As empirically noted, the total training time of APT is 18592.01 seconds, while the competitive graph pre-training model GCC takes 40161.68 seconds. The efficiency of our model is mainly due to the use of a much smaller number of carefully selected training graphs and samples at each epoch. (Detailed analysis can be found in the subsection "Training time" in Appendix H.)
>
> The time complexity of our model have been analyzed in Appendix C. The overall time complexity of APT in each batch is $O(B|V|^3 + X + B^2D + B)$, where $|V|$ is the maximal number of nodes in subgraph instances, $B$ is the batch size, $D$ is the representation dimension, and $X$ is the time complexity of backbone GNN. (Detailed analysis can be found in Appendix C.)
>
> To improve the clarity, we have now moved some crucial results to the main body and included pointer to remaining details in the appendix for further reference.
>
> >W4: The motivation of the other four property criteria is unclear, especially when they are highly correlated with the network entropy. Can we just drop the other four criteria?
>
> We appreciate the reviewer's observation. The five graph properties are provided to serve as comprehensive and informative data selection criteria. We agree that the four graph properties (i.e., density, average degree, degree variance and negative scale-free exponent) are somehow related to the network entropy, so we have conducted further empirical investigations to determine the necessity of using all five properties. The results, as reported in Table 8 of Appendix H, demonstrate the value of each individual property. Remarkably, when combining all five properties, we achieve the best performance in most cases. This emphasizes the significance of using all five properties collectively.
>
> In addition, the computational overhead of computing all these five properties is relatively low ($O(|V|)$ where $|V|$ is the maximal number of nodes in subgraph instances), and the computation is only needed once before pre-training. Hence, using all five quantities does more good than harm in our model.
>
> >W5: The authors claim in the conclusion that APT can enhance the model with a fewer number of input data, but I can't find evidence for that. Is APT only trained on fewer data than baselines?
>
> Sorry for the unclear description. APT is only trained on fewer data than baselines because of the careful selection of suitable input data. In our experiments, only 7 datasets are selected out of the available 11  (as mentioned in "Analysis of the selected graphs" in Appendix H). For each selected dataset, at most 24.92% of the samples are chosen for training. We have added detailed descriptions in the main body in the revised manuscript.
>
> >Overall weakness: Overall, the description of the proposed method could be improved. In addition, the authors claimed they are not curriculum learning but I don't think so. Time-adaptive selection strategy is just curriculum learning in a broader sense, I think easy-first is just one type of curriculum.
>
> Thanks for your constructive suggestions to improve our manuscript. Regarding the curriculum learning, thanks for reminding us that "easy-first" and "difficult-first" approaches fall under the broader category of curriculum learning. We have adjusted our previous unclear statement and improved our description in the revised manuscript.

---

> > ### Comment · Reviewer_gzMz · 2023-08-10
> > **Post rebuttal**
> >
> > The authors have addressed my concerns, and I would raise my score from 5 to 6. I would be glad to see this paper get accepted, and if so, please revise the method part to make it clear.

---

> > > ### Author Response · Authors · 2023-08-10
> > > **Thank you**
> > >
> > > Thank you for your valuable comments and timely reply. These comments help us significantly improve the quality of our work. We will make the modifications as suggested, and revise the unclear descriptions in the methodology part in our paper.

---

### Official Review · Reviewer_KR8J · 2023-07-03

**Soundness:** 3 good
**Presentation:** 2 fair
**Contribution:** 3 good
**Rating:** 6
**Confidence:** 3

**Summary:**

The paper introduced an novel graph pre-training pipeline from a data-centric view by proposing a graph data selector. The proposed approach involved sequentially feeding representative data into the model for pre-training. These representative data were determined by a data selector, which utilized prediction uncertainty and graph statistics to suggest the most relevant examples. This novel methodology offers a fresh perspective on graph pre-training and demonstrates the importance of data selection in the process.

**Strengths:**

1. The idea is novel and interesting.
2. The proposed training pipeline is tested on many tasks from different domains.

**Weaknesses:**

1. The utilization of predictive uncertainty based on pre-training loss lacks convincing evidence.

2. The proposed training pipeline requires input graphs to be processed order-by-order, which may introduce unnecessary bias and is not scalable for large graph datasets.

**Questions:**

First, there are several questions about the predictive uncertainty:

1. The loss used for graph pre-training may not effectively align with the actual loss of the downstream task [1,2]. Consequently, the predictive uncertainty based on the current definition may lack meaningful interpretation. A more promising approach would be to ensure that the uncertainty of the pretraining loss accurately reflects the difference between the pre-training loss and the downstream task loss.

2. Prediction uncertainty has been extensively explored in previous research [3,4]. While many existing methods have been proposed with ground-truth labels, it would be beneficial to have discussions (and even experiments) that shed light on the connection and distinctions between different approches of uncertainty modeling.

For experimentds

3. The experimental setups lack clarity. It is unclear whether data from all domains are used to pre-train a single model or if separate models are pre-trained for different domains.

Others

4. Please check descriptions in lines 215~217. Smaller gamma seems leading to important roles of the predictive uncertainty term.

5. Choosing subgraphs seems to be a crucial component of the pre-training algorithm. However, the paper lacks a thorough explanation or detailed descriptions of this process, which would greatly improve the understanding of this important aspect.

6. Recently, there are many related work focusing on the data-centric approaches [2,5,6], espeically for graph pre-training [2,5]. It would be great and helpful to readers if the authors could provide a brief discussion on the connections and differences between their work and these existing approaches.

Ref.

[1] Does gnn pretraining help molecular representation? NeurIPS 2022.

[2] Data-Centric Learning from Unlabeled Graphs with Diffusion Model. Arxiv.

[3] Dropout as a bayesian approximation: Representing model uncertainty in deep learning. ICML 2016.

[4] Single-model uncertainties for deep learning. NeurIPS 2019.

[5] Analyzing data-centric properties for graph contrastive learning. NeurIPS 2022.

[6] Data-centric ai: Perspectives and challenges. Arxiv.


**Limitations:**

See above.

---

> ### Author Rebuttal · Authors · 2023-08-07
>
> Dear Reviewer KR8J,
>
> Thanks for your constructive suggestions to improve our manuscript. We have followed closely the suggestions, and made clarifications and revisions accordingly. We hope the revised and newly provided content could help to further strengthen our work.
> >W1, Q1 and Q2:  Predictive uncertainty definition and its connection to existing uncertainty
>
> First, we would like to clarify that our uncertainty is introduced to enhance the graph pre-training model. Typically, graph pre-training does not consider any information from the downstream task or data, e.g., GCC, GraphCL. Therefore, the uncertainty is solely computed on pre-training data and based on the pre-training task, making the pre-training loss a natural measure for predictive uncertainty. We have added the detailed description in the revised manuscript.
>
> Second, we have presented the discussion and theoretical connection between our proposed uncertainty and existing works on uncertainty. In most existing works, uncertainty is often deﬁned on the label space, including [3, 4]. [3] quantifies uncertainty based on Orthonormal Certificates, which needs to be trained with labels. [4] leverages Bayesian learning to approximate the posterior distribution and obtain the predictive performance distribution, where the variance serves as a representation of model uncertainty. In contrast, our uncertainty is defined in the representation space. Moreover, Theorem 1 has established a theoretical connection between our uncertainty and the most conventional uncertainty defined as the cross entropy loss of an instance on the downstream classiﬁer. The theorem suggests that a smaller conventional uncertainty over all downstream classiﬁers cannot be achieved without a smaller predictive uncertainty in our work. More details can be found in lines 127-142 of our manuscript. We have highlighted this discussion and theoretical connection with a subheading in the revised version.
> >W2: Bias and scalability of order-by-order training
>
> Sorry for the confusion. We explain our order-by-order training approach and discuss the scalability as follows, which has been included in the revised version.
>
> The ordering is indeed a fundamental feature of our model, making our approach perform better than unbiased random sampling. In different stages of training process, the graph selector (equipped with predictive uncertainty and graph property) chooses the samples most needed by the current model. The experimental results present evidence of the superiority of our selection compared with unbiased random sampling in other pre-training models. Additional results are in Table 17 in Appendix H.
>
> Regarding scalability, our model is trained on a reduced number of carefully selected training graphs and samples, and achieves a training time 2.2 times faster than the competitive model GCC. In our experiments in Section 4, we carefully selected only 7 datasets out of the available 11 and performed pre-training using at most 24.92% of the samples in each selected dataset. Moreover, for each newly added dataset, our model only needs a few more training iterations to convergence, rather than being trained from scratch.
> >Q3: Experiment setup
>
> Sorry for the unclear description.  In the revised manuscript, we have highlighted the crucial setup of utilizing data from all domains to pre-train one model in Section 4.1.
> >Q4: Mistake of gamma description
>
> Thanks for pointing out this mistake. We have corrected the description in lines 215-217 to "the parameter $\gamma_t$ should be set larger so that the graph properties play a leading role. As the training phase proceeds, the graph selector gradually pays more attention to the feedback ${\phi}_{\text{uncertain}}$ from the model via a smaller value of $\gamma_t$".
> >Q5: Explanation of choosing subgraphs
>
> Thanks for pointing it out. We have included the subgraph selection process in the revised version as follows. When choosing subgraphs, we follow the practice in the existing work GCC. The process involves three sequential steps for sampling a subgraph of node $v$. (1) Random walk with restart: We start a random walk from node $v$, traversing iteratively to its neighboring nodes with the probability proportional to the edge weight. At each step, there is a positive probability for the walk to return back to the starting node $v$. (2) Subgraph induction: The random walk with restart accumulates a subset of nodes surrounding $v$. We then obtain the subgraph induced by this subset of nodes. (3) Anonymization. The induced subgraph is anonymized by relabeling its nodes in an arbitrary order.
> >Q6: Discussion of data-centric related works
>
> Thanks for reminding us of these related works. We have detailedly discussed them as follows, and included in the revised version. Data-centric AI, a recently introduced concept, emphasizes the enhancement of data quality and quantity, rather than model improvement [6]. Following-up works in graph pre-training [2, 5] exploits the data-centric idea to design data augmentation. [2] augments unlabeled graph data under the guidance of downstream, while our work focus on data selection in the pre-training phase without any information from downstream. [5] mainly focuses on the theoretical analysis of data-centric properties of data augmentation. Neither of them addresses the specific problem of data selection for pre-training, which is the objective of our proposed graph selector.
>
> The reference list of our rebuttal is the same as that of the reviewer's comment.

---

> > ### Comment · Reviewer_KR8J · 2023-08-13
> > **Thanks for the authors' response.**
> >
> > Thank you for the authors' thorough response. Most of my concerns have been addressed. I look forward to seeing the revised paper, and I hope all the key points will be appropriately covered.

---

> > > ### Author Response · Authors · 2023-08-16
> > > **Thank you**
> > >
> > > We greatly appreciate your valuable insights and prompt response. We will incorporate the suggested revisions in our paper.

---

### Official Review · Reviewer_i2Gc · 2023-07-04

**Soundness:** 3 good
**Presentation:** 3 good
**Contribution:** 3 good
**Rating:** 7
**Confidence:** 4

**Summary:**

This paper presents a data-centric framework, abbreviated as APT, for cross-domain graph pre-training. APT is composed of a graph selector and a graph pre-training model. The core idea is to select the most representative and informative data points based on the inherent properties of graphs, as well as predictive uncertainty. For the pre-training stage, when fed with the carefully selected data points, a proximal term is added to prevent catastrophic forgetting and remember all the contributions of previous input data.

**Strengths:**

1.This paper proposes that big data is not a necessity for pre-training GNNs. Instead of training on a massive amount of data, it is more reasonable to select a few suitable samples for pre-training. This approach can also reduce the amount of data and computational costs. Compared to pre-training on the entire dataset, selecting a more carefully selected subset of data for pre-training can indeed achieve better results.
2.This paper provides theoretical justification for the connection between uncertainties by establishing a provable connection between the proposed predictive uncertainty and the conventional definition of uncertainty. The predictive uncertainty is defined in the representation space, which enables the identification of more challenging samples that can benefit the model training more significantly.
3.The entire framework seems very reasonable, and the process is described clearly. From the experiments, it appears that good results have been achieved.


**Weaknesses:**

1.This paper mentions both "data-active" and "data-centric" concepts. It may be helpful to clarify the relationship between these two to avoid confusion. Maybe only using "data-active" in the paper.
2.There is no individual ablation experiment about the graph selector to demonstrate the effectiveness for each of the ﬁve graph properties.
3.There is a spelling error in the title, 'Prespective' should be corrected to 'Perspective'.


**Questions:**

Q1 Are all the Graph properties used in this paper useful when dealing with graph-inherent features? Also, what is the computational complexity since multiple properties need to be calculated?
Q2 Why did the performance of the "dd" dataset decrease after fine-tuning compared to the frozen parameters of APT in table 2?
Q3 I am slightly concerned that if all the five graph properties are useful for pre-training.
Q4 In existing works, is there any works that define model uncertainty in the representation space? And what is the difference between theirs and this work?


**Limitations:**

Yes

---

> ### Author Rebuttal · Authors · 2023-08-07
>
> Dear Reviewer i2Gc,
>
> We really appreciate your valuable comments. We have followed closely the suggestions, and made clarifications and revisions accordingly. We hope the revised and newly provided content could help to further strengthen our work.
> >W1: This paper mentions both "data-active" and "data-centric" concepts. It may be helpful to clarify the relationship between these two to avoid confusion. Maybe only using "data-active" in the paper.
>
> Thanks for your valuable suggestion. We here discuss the relation and difference of "data-active" and "data-centric" concepts. The term "data-centric" emphasizes the enhancement of data quality and quantity, rather than model improvement [1]. On the other hand, the term "data-active" in our manuscript is used to emphasize the co-evolution of data and model, rather than mere data selection before model training. Our unified pre-training framework, which co-designs both the graph selector and the pre-training model, fits in the scope of "data-active". To avoid confusion, we have consistently utilized the term "data-active" in the revised version as suggested.
>
> >W2, Q1 and Q3: W2: There is no individual ablation experiment about the graph selector to demonstrate the effectiveness for each of the ﬁve graph properties. Q1: Are all the graph properties used in this paper useful when dealing with graph-inherent features? Also, what is the computational complexity since multiple properties need to be calculated. Q3: I am slightly concerned that if all the five graph properties are useful for pre-training.
>
> The ablation studies of utilizing one graph property have been presented in Table 8 in Appendix H. The results show utilizing one graph property can still beat the best baselines in most cases, which demonstrates the effectiveness of each of the five graph properties for graph pre-training. Moreover, putting five properties together often leads to the best performance. We have added a clear pointer of the important experimental findings in the main body of our paper, making it easier for readers to locate relevant content.
>
> Regarding the computational complexity of calculating graph properties, suppose the maximal number of nodes in subgraph instances is $|V|$. We calculate average degree, degree variance, density, network entropy and scale-free exponent of each graph, which cost $O(|V|)$, $O(|V|)$, $O(1)$, $O(|V|)$ and $O(|V|)$ respectively. The overall computational complexity is $O(|V|)$. More importantly, these graph properties are only needed to compute once before pre-training. Therefore, the computational overhead is relatively low.
>
> >W3: There is a spelling error in the title, 'Prespective' should be corrected to 'Perspective'.
>
> Thank you for your meticulous attention in identifying the spelling error. We have now rectified the mistake, changing 'Prespective' to 'Perspective' in the title of the revised manuscript.
>
> >Q2: Why did the performance of the "dd" dataset decrease after fine-tuning compared to the frozen parameters of APT in table 2?
>
> Thanks for your careful reading of our paper. This phenomenon could happen since "graph pre-train and fine-tune" is an extremely complicated non-convex optimization problem. Actually the observation that fine-tuning deteriorates the performance has been made in previous work; see, e.g., [2]. We have included this discussion in the revised manuscript.
>
> >Q4: In existing works, is there any work that defines model uncertainty in the representation space? And what is the difference between theirs and this work?
>
> Thanks for your valuable insights to improve our work. After conducting a comprehensive literature review, we find that the majority of existing works define uncertainty in the label space, such as taking the uncertainty as the confidence level about the prediction [5-8]. Only a few works define uncertainty in the representation space [3,4]. In [3], uncertainty is measured based on the representations of an instance's nearest neighbors with the same label. However, this approach requires access to the label information of the neighbors, and thus cannot be adapted in pre-training with unlabeled data. [4] introduces a pretext task for training a model of uncertainty over the learned representations, but this method assumes a well-pre-trained model is already available. Such a post processing manner is not applicable to our scenario, because we need an uncertainty that can guide the selection of pre-training data during pre-training rather than after pre-training. We have included a detailed discussion of these related works in the revised manuscript.
>
> [1] Data-centric Artificial Intelligence: A Survey. Arxiv.
>
> [2] Fine-tuning can distort pre-trained features and underperform out-of-distribution. ICLR.
>
> [3] Density estimation in representation space to predict model uncertainty. EDSMLS.
>
> [4] A Simple Framework for Uncertainty in Contrastive Learning. Arxiv.
>
> [5] A Survey of Uncertainty in Deep Neural Networks. Arxiv.
>
> [6] Simple and Principled Uncertainty Estimation with Deterministic Deep Learning via Distance Awareness. NeurIPS.
>
> [7] Dropout as a bayesian approximation: Representing model uncertainty in deep learning. ICML.
>
> [8] Single-model uncertainties for deep learning. NeurIPS.

---

> > ### Comment · Reviewer_i2Gc · 2023-08-18
> >
> > Thanks for the rebuttal. I notice that there have been some recent papers about data-centric GNNs. I suggest to keep the related work and discussion up-to-date if accepted. I will raise my score to 7.

---

> > > ### Author Response · Authors · 2023-08-20
> > > **Thank you**
> > >
> > > Thank you for your support of our work! We sincerely value your insightful suggestions, and we will incorporate the suggested revisions into our paper.

---

### Official Review · Reviewer_16eq · 2023-07-08

**Soundness:** 3 good
**Presentation:** 3 good
**Contribution:** 2 fair
**Rating:** 6
**Confidence:** 4

**Summary:**

This paper proposes a novel approach to pre-training graph neural networks (GNNs) called the data-active graph pre-training (APT) framework. This framework introduces a unique method that involves a graph selector and a pre-training model. These two components work together in a progressive and iterative way to choose the most representative and instructive data points for pre-training.

**Strengths:**

1. The key insight of the paper is that using fewer, but carefully selected data can lead to better downstream performance compared to using a massive amount of input data. This approach challenges the common practice in machine learning of using large datasets for pre-training, a phenomenon the authors refer to as the "curse of big data" in graph pre-training.

2. The paper is well-motivated, well-written and easy to understand.

3. The experimental results demonstrate that the proposed method outperformed GCC, which is the previous SOTA and a direct ablation to the proposed APT framework.

**Weaknesses:**

1. The proposed method is incremental. The proposed “predictive uncertainty” is adapted from curriculum learning, and “graph properties” are common statistics.

2. The proposed methods are not able to handle node features. It only provides structural information which does not align with most real-world scenarios. As a result, performance is not comparable with a supervised model trained on node features.

3. The node classification performance on homophily graphs (which is an important family of graphs) are much lower than ProNE. The improvement in graph classification tasks seems limited.

4. Selection based on graph properties might cause potential test data leakage because the graph properties are selected according to test performance (as shown in Figure 2).

**Questions:**

In line 67, section 2, the authors present an example of a triangle to show what kind of knowledge the pre-trained model can learn. However, the proposed subgraph instance discrimination task can break down that knowledge. For example, ego-graph sampling can break down the triangle. Can the authors provide some insights or justification for this issue?

**Limitations:**

Yes

---

> ### Author Rebuttal · Authors · 2023-08-07
>
> Dear Reviewer 16eq,
>
> We really appreciate your insightful comments. We have followed closely the suggestions, and made clarifications and revisions accordingly. We hope the revised and newly provided content could help to further strengthen our work. Below is a point-by-point response.
> >W1: predictive uncertainty from curriculum learning
>
> We appreciate the reviewer's careful evaluation of our proposed method. We elucidate three significant differences between curriculum learning and our uncertainty: (1) The uncertainty in most curriculum learning is built on label space for supervised learning [1-4], while we define the uncertainty in representation space and further establish its theoretical connection with the label space. (2) In curriculum learning, samples are individually selected based on uncertainty without considering their joint distribution [1,2]. However, in graph pre-training, it is crucial to ensure that the chosen samples conform to a joint distribution that reflects data diversity and topological structures of real-world graphs. This is achieved through the incorporation of graph properties that characterize the statistics of joint distribution of samples. (3) Most traditional curriculum learning determines the order of learning samples before model training [1,2,5,6]. In our model, however, the uncertainty changes during different stages of model training. Thus, the next training sample is not determined util the end of the previous iteration. Therefore, our method has substantial difference from curriculum learning.
> > W2: Transferability of node features
>
> We appreciate the insightful point. Here we focus on the transferability of graph structures. It is also a common practice to transfer structure rather than node features in related research areas including graph pre-training [10], transfer learning [11] and graph domain adaptation [7-9]. This is because, in most cases, node features of different graphs do not overlap or have little overlap, making them not necessarily transferable across pre-training and downstream data. For instance, when nodes in different graphs represent different types of entities, it renders the node features completely irrelevant. Even when nodes are of the same entity type across graphs, the dimensions/meanings of node features can vary significantly, leading to misalignment. After all, the transferability of node features in graph pre-training is a largely unexplored area, and needs further research attention.
> >W3: Performance on homophily graphs and graph classification
>
> Thank you for bringing up this point, and we here delve into the specific cases in detail.
>
> Regarding the performance on homophily graphs, the design of ProNE is based on the homophily assumption that neighboring nodes share the same class. What's more, ProNE requires re-training on different graphs, making it non-transferable. On the contrary, we target at a general, transferable model free from any specific assumptions on graphs, and thus applicable to various settings, including both homophily and heterophily graphs. This could explain the better performance of ProNE on homophily graphs, and the superiority of our model in most other cases (particularly in situations where ProNE fails).  We have included this discussion in the revised manuscript.
>
> Regarding graph classification performance. Admittedly, our model shows modest improvements in this aspect. Nevertheless, it is valid to emphasize the substantial efficiency gain: APT obtains comparable performance with training time 2.2 times shorter than the most competitive baseline GCC. This efficiency advantage holds practical significance and offers real-world application benefits.
> >W4: Potential data leakage
>
> Sorry for the misunderstanding. To clarify, each downstream dataset is split into a training set and a test set. When selecting graph properties, we only refer to the performance on the **training set** of downstream dataset, including the results in Figure 2(b). We remark that the test set of downstream dataset is completely excluded during model training and graph property selection, preventing any form of data leakage. We have revised the manuscript to avoid any ambiguity.
> >Q1: Sampling in subgraph instance discrimination task
>
> Thanks for the insightful comments. In the subgraph instance discrimination task, one subgraph is defined as the neighborhood of a node. Such neighborhood typically has complex structure, comprising of repeated occurrences of various simple structural patterns (e.g., open triangles, closed triangles). Therefore, although the ego-graph sampling may break some simple structural patterns, most information in the complex neighborhood is still preserved. We have provided the discussions in the revised version.
>
> [1] Curriculum learning: A survey. IJCV.
>
> [2] A survey on curriculum learning. TPAMI.
>
> [3] When do curricula work? ICLR.
>
> [4] Dynamically composing domain-data selection with clean-data selection by "co-curricular learning" for neural machine translation. ACL.
>
> [5] On the power of curriculum learning in training deep networks. ICML.
>
> [6] Curriculum learning by transfer learning: Theory and experiments with deep networks. ICML.
>
> [7] Graph domain adaptation: A generative view. ArXiv.
>
> [8] Graph adaptive knowledge transfer for unsupervised domain adaptation. ECCV.
>
> [9] Graph Domain Adaptation via Theory-Grounded Spectral Regularization. ICLR.
>
> [10] GCC: Graph contrastive coding for graph neural network pre-training. SIGKDD.
>
> [11] Transfer learning of graph neural networks with ego-graph information maximization. NeurIPS.

---

> > ### Comment · Reviewer_16eq · 2023-08-19
> > **Thanks for the rebuttal**
> >
> > Thank you for the detailed reply and additional experiments. Most of my concerns have been addressed and I increased my score to 6.

---

> > > ### Author Response · Authors · 2023-08-20
> > > **Thank you**
> > >
> > > Thank you for your support for our work! We greatly appreciate your valuable suggestions, and we will incorporate the suggested revisions into our paper.

---

### Decision · Program_Chairs · 2023-09-21

**Decision:**

Accept (poster)

**Comment:**

This submission offers a refreshing perspective on the realm of pre-training GNNs. The reviewers have arrived at a consensus appreciating multiple facets of the paper:

* The central proposition of the paper stands out — using fewer, strategically selected data can lead to enhanced downstream performance, challenging the standard "more is better" approach. Reviewers commend the clarity of the manuscript. The paper is well-articulated and coherent, making it accessible to readers.

* Experimentally, the paper showcases significant prowess. The results indicate that the proposed method surpasses the previous state-of-the-art. Another commendable aspect is the paper's theoretical justification. The work establishes a link between the introduced predictive uncertainty and traditional definitions of uncertainty, enhancing its credibility. By defining the predictive uncertainty in the representation space, the paper offers a novel mechanism to pinpoint samples that can benefit the training.

It is evident that the submission offers a novel, well-founded, and rigorously tested methodology that has the potential to reshape thinking and practices in pre-training GNNs. Given its strengths, both in theoretical grounding and empirical validation, the manuscript is considered fit for acceptance at NeurIPS.